# On Infinite-Width Hypernetworks

**Etai Littwin**[*]
School of Computer Science
Tel Aviv University
Tel Aviv, Israel
etai.littwin@gmail.com

**Tomer Galanti**[*]
School of Computer Science
Tel Aviv University
Tel Aviv, Israel
tomerga2@tauex.tau.ac.il

**Lior Wolf**
School of Computer Science
Tel Aviv University
Tel Aviv, Israel
wolf@cs.tau.ac.il

**Greg Yang**
Microsoft Research AI
gregyang@microsoft.com

## Abstract

*Hypernetworks* are architectures that produce the weights of a task-specific *primary network*. A notable application of hypernetworks in the recent literature involves learning to output functional representations. In these scenarios, the hypernetwork learns a representation corresponding to the weights of a shallow MLP, which typically encodes shape or image information. While such representations have seen considerable success in practice, they remain lacking in the theoretical guarantees in the wide regime of the standard architectures. In this work, we study wide over-parameterized hypernetworks. We show that unlike typical architectures, infinitely wide hypernetworks do not guarantee convergence to a global minima under gradient descent. We further show that convexity can be achieved by increasing the dimensionality of the hypernetwork's output, to represent wide MLPs. In the dually infinite-width regime, we identify the functional priors of these architectures by deriving their corresponding GP and NTK kernels, the latter of which we refer to as the *hyperkernel*. As part of this study, we make a mathematical contribution by deriving tight bounds on high order Taylor expansion terms of standard fully connected ReLU networks.

## 1  Introduction

In this work, we analyze the training dynamics of over-parameterized meta networks, which are networks that output the weights of other networks, often referred to as *hypernetworks*. In the typical framework, a function $h$ involves two networks, $f$ and $g$. The *hypernetwork* $f$ takes the input $x$ (typically an image) and returns the weights of the *primary network*, $g$, which then takes the input $z$ and returns the output of $h$.

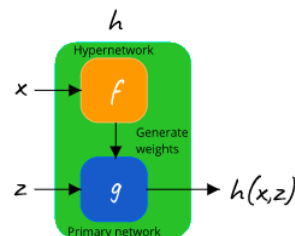

The literature of hypernetworks is roughly divided into two main categories. In the functional representation literature [22, 32, 38, 29, 16] the input to the hypernetwork $f$ is typically an image. For shape reconstruction tasks, the network $g$ represents the shape via a signed distance field, where the input are coordinates in 3D space. In image completion tasks, the inputs to

---

[*]Equal Contribution

$g$ are image coordinates, and the output is the corresponding pixel intensity. In these settings, $f$ is typically a large network and $g$ is typically a shallow fully connected network.

In the second category [4, 23, 35, 43], hypernetworks are typically used for hyper-parameter search, where $x$ is being treated as a hyperparameter descriptor and is optimized alongside with the network's weights. In this paper, we consider models corresponding to the first group of methods.

Following a prominent thread in the recent literature, our study takes place in the regime of wide networks. [11] recently showed that, when the width of the network approaches infinity, the gradient-descent training dynamics of a fully connected network $f$ can be characterized by a kernel, called the *Neural Tangent Kernel* (or NTK for short). In other words, as the width of each layer approaches infinity, provided with proper scaling and initialization of the weights, it holds that:

$$\frac{\partial f(x; w)}{\partial w} \cdot \frac{\partial^\top f(x'; w)}{\partial w} \to \Theta^f(x, x') \tag{1}$$

as the width, $n$ of $f$ tends to infinity. Here, $w$ are the weights of the network $f(x; w)$. As shown in [11], as the width tends to infinity, when minimizing the squared loss using gradient descent, the evolution through time of the function computed by the network follows the dynamics of kernel gradient descent with kernel $\Theta^f$. To prove this phenomenon, various papers [20, 3, 2] introduce a Taylor expansion of the network output around the point of initialization and consider its values. It is shown that the first-order term is deterministic during the SGD optimization and the higher-order terms converge to zero as the width $n$ tends to infinity.

A natural question that arises when considering hypernetworks is whether a similar "wide" regime exists, where trained and untrained networks may be functionally approximated by kernels. If so, since this architecture involves two networks, the "wide" regime needs a more refined definition, taking into account both networks.

Our contributions:

1. We show that infinitely wide hypernetworks can induce highly non-convex training dynamics under gradient descent. The complexity of the optimization problem is highly dependent on the architecture of the primary network $g$, which may considerably impair the trainability of the architecture if not defined appropriately.

2. However, when the widths of both the hypernetwork $f$ and the primary network $g$ tend to infinity, the optimization dynamics of the hypernetwork simplifies, and its neural tangent kernel (which we call the *hyperkernel*) has a well defined infinite-width limit governing the network evolution.

3. We verify our theory empirically and also demonstrate the utility of this *hyperkernel* on several functional representation tasks. Consistent with prior observations on kernel methods, the hypernetwork induced kernels also outperforms a trained hypernetwork when training data is small.

4. We make a technical contribution by deriving asymptotically tight bounds on high order Taylor expansion terms in ReLU MLPs. Our result partially settles a conjecture posed in [6] regarding the asymptotic behavior of general correlation functions.

## 1.1 Related Works

**Hypernetworks** Hypernetworks were first introduced under this name in [8], are networks that generate the weights of a second *primary* network that computes the actual task. However, the idea of having one network predict the weights of another was proposed earlier and has reemerged multiple times [15, 28, 13]. The tool can naturally be applied for image representations tasks. In [22], they applied hypernetworks for 3D shape reconstruction from a single image. In [32] hypernetworks were shown to be useful for learning shared image representations. Hypernetworks were also shown to be effective in non-image domains. For instance, hypernetworks achieve state of the art results on the task of decoding error correcting codes [24].

Several publications consider a different framework, in which, the inputs $x$ of the hypernetwork are optimized alongside to the weights of the hypernetwork. In this setting, hypernetworks were recently used for continuous learning by [35]. Hypernetworks can be efficiently used for neural architecture search, as was demonstrated by [4, 43], where a feedforward regression (with network $f$) replaces

direct gradient-based learning of the weights of the primary network while its architecture is being explored. Lorraine et al. applied hypernetworks for hyperparameters selection [23].

Despite their success and increasing prominence, little theoretical work was done in order to better understand hypernetworks and their behavior. A recent paper [12] studies the role of multiplicative interaction within a unifying framework to describe a range of classical and modern neural network architectural motifs, such as gating, attention layers, hypernetworks, and dynamic convolutions amongst others. It is shown that standard neural networks are a strict subset of neural networks with multiplicative interactions. In [7] the authors theoretically study the modular properties of hypernetworks. In particular, they show that compared to standard embedding methods, hypernetworks are exponentially more expressive when the primary network is of small complexity. In this work, we provide a complementary perspective and show that a shallow primary network is a requirement for successful training. [5] showed that applying standard initializations on a hypernetwork produces sub-optimal initialization of the primary network. A principled technique for weight initialization in hypernetworks is then developed.

**Gaussian Processes and Neural Tangent Kernel**   The connection between infinitely wide neural networks, Gaussian processes and kernel methods, has been the focus of many recent papers [11, 19, 39, 41, 31, 30, 37, 36, 25]. Empirical support has demonstrated the power of CNTK (convolutional neural tangent kernel) on popular datasets, demonstrating new state of the art results for kernel methods [1, 42]. [21] showed that ReLU ResNets [10] can have NTK convergence occur even when depth and width simultaneously tend to infinity, provided proper initialization. In this work, we extend the kernel analysis of networks to hypernetworks, and characterize the regime in which the kernels converge and training dynamics simplify.

## 2   Setup

In this section, we introduce the setting of the analysis considered in this paper. We begin by defining fully connected neural networks and hypernetworks in the context of the NTK framework.

**Neural networks**   In the NTK framework, a fully connected neural network, $f(x; w) = y^L(x)$, is defined in the following manner:

$$\begin{cases} y^l(x) = \sqrt{\frac{1}{n_{l-1}}} W^l q^{l-1}(x) \\ q^l(x) = \sqrt{2} \cdot \sigma(y^l(x)) \end{cases} \quad \text{and } q^0(x) = x \,, \tag{2}$$

where $\sigma : \mathbb{R} \to \mathbb{R}$ is the activation function of $f$. Throughout the paper, we specifically take $\sigma$ to be a piece-wise linear function with a finite number of pieces (e.g., the ReLU activation $\text{ReLU}(x) := \max(0, x)$ and the Leaky ReLU activation $\text{ReLU}_\alpha(x) = \begin{cases} x & \text{if } x \geq 0 \\ \alpha x & \text{if } x < 0 \end{cases}$). The weight matrices $W^l \in \mathbb{R}^{n_l \times n_{l-1}}$ are trainable variables, initialized independently according to a standard normal distribution, $W^l_{i,j} \sim \mathcal{N}(0, 1)$. The width of $f$ is denoted by $n := \min(n_1, \ldots, n_{L-1})$. The parameters $w$ are aggregated as a long vector $w = (vec(W^1), \ldots, vec(W^L))$. The coefficients $\sqrt{1/n_{l-1}}$ serve for normalizing the activations of each layer. This parametrization is nonstandard, and we will refer to it as the NTK parameterization. It has already been employed in several recent works [14, 34, 26]. For simplicity, in many cases, we will omit to specify the weights $w$ associated with our model.

**Hypernetworks**   Given the input tuple $u = (x, z) \in \mathbb{R}^{n_0 + m_0}$, we consider models of the form: $h(u; w) := g(z; f(x; w))$, where $f(x; w)$ and $g(z; v)$ are two neural network architectures with depth $L$ and $H$ respectively. The function $f(x; w)$ referred to as *hypernetwork*, takes the input $x$ and computes the weights $v = f(x; w)$ of a second neural network $g(z; v)$, referred as the *primary network*, which is assumed to output a scalar. As before, the variable $w$ stands for a vector of trainable parameters ($v$ is not trained directly and is given by $f$).

We parameterize the primary network $g(z; v) = g^H(z; v)$ as follows:

$$\begin{cases} g^l(z; v) = \sqrt{\frac{1}{m_{l-1}}} V^l \cdot a^{l-1}(z; v) \\ a^l(z; v) = \sqrt{2} \cdot \phi(g^l(z; v)) \end{cases} \quad \text{and } a^0(z) = z \tag{3}$$

Here, the weights of the primary network $V^l(x) \in \mathbb{R}^{m_l \times m_{l-1}}$ are given in a concatenated vector form by the output of the hypernetwork $f(x; w) = v = (vec(V^1), \ldots, vec(V^H))$. The output dimension of the hypernetwork $f$ is therefore $n_L = \sum_{i=1}^{H} m_i \cdot m_{i-1}$. We denote by $f^d(x; w) := V^d(x; w) := V^d$ the $d$'th output matrix of $f(x; w)$. The width of $g$ is denoted by $m := \min(m_1, \ldots, m_{H-1})$. The function $\phi$ is an element-wise continuously differentiable function or a piece-wise linear function with a finite number of pieces.

**Optimization**    Let $S = \{(u_i, y_i)\}_{i=1}^{N}$, where $u_i = (x_i, z_i)$ be some dataset and let $\ell(a, b) := |a - b|^p$ be the $\ell^p$-loss function. For a given hypernetwork $h(u; w)$, we are interested in selecting the parameters $w$ that minimize the empirical risk:

$$c(w) := \sum_{i=1}^{N} \ell(h(u_i; w), y_i) \tag{4}$$

For simplicity, oftentimes we will simply write $\ell_i(a) := \ell(a, y_i)$ and $h_i(w) := h(u_i) := h(u_i; w)$, depending on the context. In order to minimize the empirical error $c(w)$, we consider the SGD method with learning rate $\mu > 0$ and step of the form $w_{t+1} \leftarrow w_t - \mu \nabla_w \ell_{j_t}(h_{j_t}(w_t))$ for some index $j_t \sim U[N]$ that is selected uniformly at random for the $t$'th iteration. A continuous version of the GD method is the gradient flow method, in which $\dot{w} = -\mu \nabla_w c(w)$. In recent works [14, 20, 1, 34], the optimization dynamics of the gradient method for standard fully-connected neural networks was analyzed, as the network width tends to infinity. In our work, since hypernetworks consist of two interacting neural networks, there are multiple ways in which the size can tend to infinity. We consider two cases: (i) the width of $f$ tends to infinity and that of $g$ is fixed and (ii) the width of both $f$ and $g$ tend to infinity.

## 3  Dynamics of Hypernetworks

**Infinitely wide $f$ without infinitely wide $g$ induces non-convex optimization**    In the NTK literature, it is common to adopt a functional view of the network evolution by analyzing the dynamics of the output of the network along with the cost, typically a convex function, as a function of that output. In the hypernetwork case, this presents us with two possible viewpoints of the same optimization problem of $h(u) = g(z; f(x))$. On one hand, since only the hypernetwork $f$ contains the trainable parameters, we can view the optimization of $h$ under the loss $\ell$ as training of $f$ under the loss $\ell \circ g$. The classical NTK theory would imply that $f$ evolves linearly when its width tends to infinity, but because $\ell \circ g$ is in general not convex anymore, even when $\ell$ originally is, an infinitely wide $f$ without an infinitely wide $g$ does not guarantee convergence to a global optimum. In what follows, we make this point precise by characterising how nonlinear the dynamics becomes in terms of the depth of $g$.

After a single stochastic gradient descent step with learning rate $\mu$, the hypernetwork output for example $i$ is given by $h_i(w - \mu \nabla_w \ell_j)$. When computing the Taylor approximation around $w$ with respect to the function $h$ at the point $w' = w - \mu \nabla_w \ell_j$, it holds that:

$$h_i(w - \mu \nabla_w \ell_j) = \sum_{r=0}^{\infty} \frac{1}{r!} \langle \nabla_w^{(r)} h_i, (-\mu \nabla_w \ell_j)^r \rangle = \sum_{r=0}^{\infty} \frac{1}{r!} \left( -\mu \frac{\partial \ell_j}{\partial h_j} \right)^r \cdot \mathcal{K}_{i,j}^{(r)} \tag{5}$$

where $\mathcal{K}_{i,j}^{(r)} := \langle \nabla_w^{(r)} h_i, (\nabla_w h_j)^r \rangle$, and $\nabla_w^{(r)} h_i$ is the $r$ tensor that holds the $r$'th derivative of the output $h_i$. The terms $\langle \nabla_w^{(r)} h_i, (-\mu \nabla_w \ell_j)^r \rangle$ are the multivariate extensions of the Taylor expansion terms $h_i^{(r)}(w)(w' - w)^r$, and take the general form of correlation functions as introduced in Eq. 5 in the appendix. This equation holds for neural networks with smooth activation functions (including hypernetworks), and holds in approximation for piece-wise linear activation functions.

Previous works have shown that, if $h$ is a wide fully connected network, the first order term $(r = 1)$ converges to the NTK, while higher order terms $(r > 1)$ scale like $\mathcal{O}(1/\sqrt{n})$ [20, 6]. Hence, for large widths and small learning rates, these higher order terms vanish, and the loss surface appears deterministic and linear at initialization, and remains so during training.

However, the situation is more complex for hypernetworks. As shown in the following theorem, for infinitely wide hypernetworks and finite primary network, the behaviour depends on the depth and width of the generated primary network. Specifically, when the primary network is deep and narrow, the higher order terms in Eq. 5 may not vanish, and parameter dynamics can be highly non-convex.

**Theorem 1** (Higher order terms for hypernetworks). *Let $h(u) = g(z; f(x))$ for a hypernetwork $f$ and an primary network $g$. Then, we have:*

$$\mathcal{K}_{i,j}^{(r)} \sim \begin{cases} n^{H-r} & \text{if } r > H \\ 1 & \text{otherwise.} \end{cases} \tag{6}$$

Thm. 1 illustrates the effect of the depth of the primary network $g$ on the evolution of the output $h$. The larger $H$ is, the more non-linear the evolution is, even when $f$ is infinitely wide. Indeed, we observe empirically that when $f$ is wide and kept fixed, a deeper $g$ incurs slower training, and lower overall test performance as illustrated in Fig. 2.

As a special case of this theorem, when taking $H = 1$, we can also derive the asymptotic behaviour of $\mathcal{K}_{i,j}^{(r)} \sim n^{1-r}$ for a neural network $h$. This provides a tighter bound than the previously conjectured $\mathcal{O}(1/n)$ upper bound [6]. The following remark is a consequence of this result and is validated in the supplementary material.

**Remark 1.** *The $r$'th order term of the Taylor expansion in Eq. 5 is of order $\mathcal{O}(\frac{\mu^r}{r! \cdot n^{r-1}})$ instead of the previously postulated $\mathcal{O}(\frac{\mu^r}{r! \cdot n})$. Therefore, it is evident that for any choice $\mu = o(\sqrt{n})$, all of the high order terms tend to zero as $n \to \infty$. This is opposed the previous bound, which guarantees that all of the high order terms tend to zero as $n \to \infty$ only when $\mu$ is constant.*

## 4  Dually Infinite Hypernetworks

It has been shown by [11, 19] that NNGPs and neural tangent kernels fully characterise the training dynamics of infinitely wide networks. As a result, in various publications [21, 9], these kernels are being treated as functional priors of neural networks. In the previous section, we have shown that the Taylor expansion of the hypernetwork is non-linear when the size of the primary network is finite. In this section, we consider the case when *both* hyper and primary networks are infinitely wide, with the intention of gaining insight into the functional prior of wide hypernetworks. For this purpose, we draw a formal correspondence between infinitely wide hypernetworks and GPs, and use this connection to derive the corresponding neural tangent kernel.

### 4.1  The NNGP kernel

Previous work have shown the equivalence between popular architectures, and Gaussian processes, when the width of the architecture tends to infinity. This equivalence has sparked renewed interest in kernel methods, through the corresponding NNGP kernel, and the Neural Tangent Kernel (NTK) induced by the architecture, which fully characterise the training dynamics of infinitely wide networks. This equivalence has recently been unified to encompass most architectures which use a pre-defined set of generic computational blocks [39, 40]. Hypernetworks represent a different class of neural networks where the parameters contain randomly initialized matrices except the last layer whose parameters are aggregated as a rank 3 tensor. All of the matrices/tensors dimensions tend to infinity. This means the results of [39, 40] do not apply to hypernetworks. Nevertheless, by considering sequential limit taking, where we take the limit of the width of $f$ ahead of the width of $g$, we show the output of $f$ achieves a GP behaviour, essentially feeding $g$ with Gaussian distributed weights with adaptive variances. A formal argument is presented in the following theorem.

**Theorem 2** (Hypernetworks as GPs). *Let $h(u) = g(z; f(x))$ be a hypernetwork. For any pair of inputs $u = (x, z)$ and $u' = (x', z')$, let $\Sigma^0(z, z') = \frac{z^\top z'}{m_0}, S^0(x, x') = \frac{x^\top x'}{n_0}$. Then, it holds for any unit $i$ in layer $0 < l \leq H$ of the primary network:*

$$g_i^l(z; f(x)) \xrightarrow{d} \mathcal{G}_i^l(u) \tag{7}$$

*as $m, n \to \infty$ sequentially. Here, $\{\mathcal{G}_i^l(u)\}_{i=1}^{m_l}$ are independent Gaussian processes, such that, $(\mathcal{G}_i^l(u), \mathcal{G}_i^l(u')) \sim \mathcal{N}(0, \Lambda^l(u, u'))$ defined by the following recursion:*

$$\Lambda^{l+1}(u, u') = \begin{pmatrix} \Sigma^l(u, u) & \Sigma^l(u', u) \\ \Sigma^l(u, u') & \Sigma^l(u', u') \end{pmatrix} \odot \begin{pmatrix} S^L(x, x) & S^L(x', x) \\ S^L(x, x') & S^L(x', x') \end{pmatrix} \tag{8}$$

$$\Sigma^l(u, u') = 2 \mathop{\mathbb{E}}_{(u,v) \sim \mathcal{N}(0, \Lambda^l)} [\sigma(u) \cdot \sigma(v)] \tag{9}$$

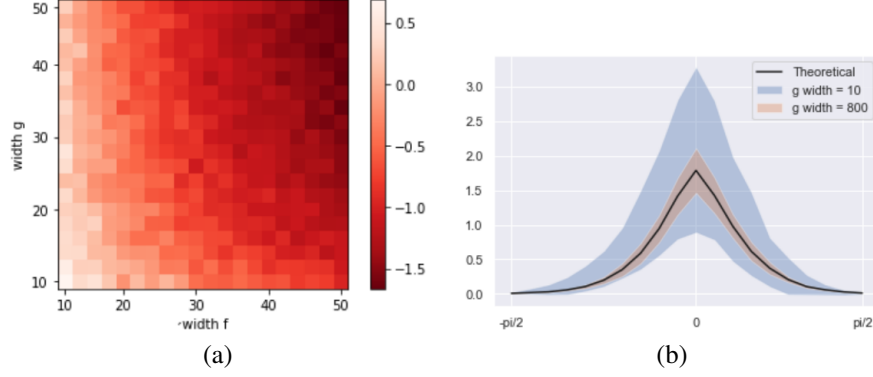

(a)                                                        (b)

Figure 1: **Convergence to the hyperkernel.** **(a)** Empirical variance of kernel values in log-scale for a single entry for varying width $f$ and $g$. Variance of the kernel converges to zero only when the widths of $f$ and $g$ both increase. **(b)** Empirical kernel value for $z = (1,0)$, $z' = (\cos(\theta), \sin(\theta))$, and $x = x' = (1,0)$ for different values of $\theta \in [-\frac{\pi}{2}, \frac{\pi}{2}]$. Convergence to a deterministic kernel is observed only when both $f$ and $g$ are wide.

*where $S^L(x,x')$ is defined recursively:*

$$S^l(x,x') = 2 \mathop{\mathbb{E}}_{(u,v)\sim\mathcal{N}(0,\Gamma^l)}[\sigma(u)\cdot\sigma(v)] \text{ and } \Gamma^l(x,x') = \begin{pmatrix} S^l(x,x) & S^l(x',x) \\ S^l(x,x') & S^l(x',x') \end{pmatrix} \qquad (10)$$

In other words, the NNGP kernel, governing the behaviour of wide untrained hypernetworks, is given by the Hadamard product of the GP kernels of $f$ and $g$ (see Eq. 8).

As a consequence of the above theorem, we observe that the NNGP kernel of $h$ at each layer, $\Lambda^l(u,u')$, is simply a function of $\Sigma^0(z,z')$, $S^0(x,x')$.

**Corollary 1.** *Let $h(u) = g(z; f(x))$ be a hypernetwork. For any $0 < l \leq H$, there exists a function $\mathcal{F}^l$, such that, for all pairs of inputs $u = (x,z)$ and $u' = (x',z')$, it holds that:*

$$\Lambda^H(u,u') = \mathcal{F}\left(\Sigma^0(z,z'), S^0(x,x')\right) \qquad (11)$$

The factorization of the NNGP kernel into a function of $\Sigma^0(z,z')$ and $S^0(x,x')$ provides a convenient way to explicitly encode useful invariances into the kernel.

As an example, in the following remark, we investigate the behaviour of the NNGP kernel of $h$, when the inputs $z$ are preprocessed random random Fourier features as suggested by [27, 33].

**Remark 2.** *Let $p(z) = [\cos(W_i^1 z + b_i^1)]_{i=1}^k$ be a Fourier features preprocessing, where $W_{i,j}^1 \sim \mathcal{N}(0,1)$ and biases $b_i \sim U[-\pi,\pi]$. Let $h(u) = g(p(z); f(x))$ be a hypernetwork, with $z$ preprocessed according to $p$. Let $u = (x,z)$ and $u' = (x',z')$ be two pairs of inputs. Then, $\Lambda^l(u,u')$ is a function of $\exp[-\|z - z'\|_2^2/2]$ and $S^L(x,x')$.*

The above remark shows that for any given inputs $x, x'$, the NNGP kernel depends on $z, z'$ only through the distance between $z$ and $z'$, which has been shown to be especially useful in implicit neural representation [33].

We next derive the corresponding neural tangent kernel of hypernetworks, referred to as hyperkernels.

## 4.2 The Hyperkernel

Recall the definition of the NTK as the infinite width limit of the Jacobian inner product given by:

$$\mathcal{K}^h(u,u') = \frac{\partial h(u)}{\partial w} \cdot \frac{\partial^\top h(u')}{\partial w} = \frac{\partial g(z; f(x))}{\partial f(x)} \cdot \mathcal{K}^f(x,x') \cdot \frac{\partial^\top g(z'; f(x'))}{\partial f(x')} \qquad (12)$$

where $\mathcal{K}^f(x,x') := \frac{\partial f(x)}{\partial w} \cdot \frac{\partial^\top f(x')}{\partial w}$ and $\mathcal{K}^g(u,u') := \frac{\partial g(z; f(x))}{\partial f(x)} \cdot \frac{\partial^\top g(z'; f(x'))}{\partial f(x')}$. In the following theorem we show that $\mathcal{K}^h(u,u')$ converges in probability at initialization to a limiting kernel in the

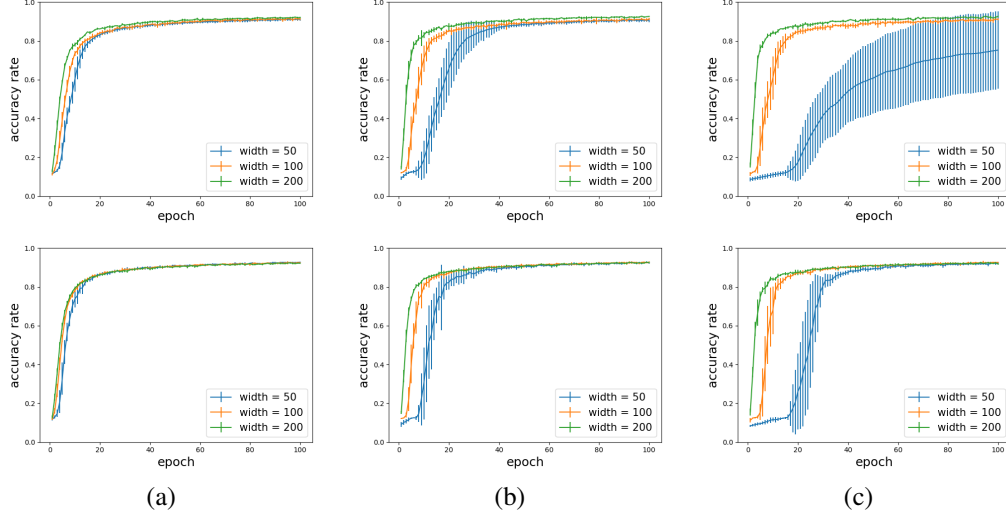

Figure 2: **A hypernetwork with a wider and *shallower* primary network** $g$ **trains faster and achieves better test performance** on the MNIST (**top**) and CIFAR10 (**bottom**) rotations prediction task. We fix the hypernetwork $f$ and the depth of $g$ at (**a**) 3, (**b**) 6 and (**c**) 8, while varying the width $m$ of $g$. The x-axis specifies the epoch and the y-axis the accuracy at test time.

sequentially infinite width limit of $f$ and $g$, denoted by $\Theta^h(u, u')$. Furthermore, we show that the hyperkernel is decomposed to the Hadamard product between the kernels corresponding to $f$ and $g$. In addition, we show that the derivative of the hyperkernel with respect to time tends to zero at initialization.

**Theorem 3** (Hyperkernel decomposition and convergence at initialization). *Let* $h(u; w) = g(z; f(x; w))$ *be a hypernetwork. Then,*

$$\mathcal{K}^h(u, u') \xrightarrow{p} \Theta^h(u, u') \tag{13}$$

*where:*

$$\Theta^h(u, u') = \Theta^f(x, x') \cdot \Theta^g(u, u', S^L(x, x')) \tag{14}$$

*such that:*

$$\mathcal{K}^f(x, x') \xrightarrow{p} \Theta^f(x, x') \cdot I \text{ and } \mathcal{K}^g(u, u') \xrightarrow{p} \Theta^g(u, u', S^L(x, x')) \tag{15}$$

*moreover, if* $w$ *evolves throughout gradient flow, we have:*

$$\left. \frac{\partial \mathcal{K}^h(u, u')}{\partial t} \right|_{t=0} \xrightarrow{p} 0 \tag{16}$$

*where the limits are taken with respect to* $m, n \to \infty$ *sequentially.*

As a consequence of Thm. 3, when applying a Fourier features preprocessing to $z$, one obtains that $\Theta^g(u, u')$ becomes shift invariant.

**Remark 3.** *Let* $p(z)$ *be as in Remark 2. Let* $h(u) = g(p(z); f(x))$ *be a hypernetwork, where* $z$ *is preprocessed according to* $p$. *Let* $u = (x, z)$ *and* $u' = (x', z')$ *be two pairs of inputs. Then,* $\Theta^g(u, u')$ *is a function of* $\exp[-\|z - z'\|_2^2 / 2]$ *and* $S^0(x, x')$.

Note that $\Theta^f$ is the standard limiting NTK of $f$ and depends only on the inputs $\{x_i\}_{i=1}^N$. However from Eq. 8, the term $\Theta^g$ requires the computation of the NNGP kernel of $f$ in advance in order to compute the terms $\{\Sigma^l, \dot{\Sigma}^l\}$. This form provides an intuitive factorization of the hyperkernel into a term $\Theta^f$ which depends on the meta function and data, and $\Theta^g$ which can be though of as a conditional term.

## 5    Experiments

Our experiments are divided into two main parts. In the first part, we validate the ideas presented in our theoretical analysis and study the effect of the width and depth of $g$ on the optimization of a hypernetwork. In the second part, we evaluate the performance of the NNGP and NTK kernels on image representation tasks. For further implementation details on all experiments see Appendix A.

| | Representation | | | | Inpainting | | | |
|---|---|---|---|---|---|---|---|---|
| $N$ | 50 | 100 | 200 | 500 | 50 | 100 | 200 | 500 |
| HK | 0.055 | 0.050 | 0.043 | 0.032 | 0.057 | 0.051 | 0.047 | 0.038 |
| NNGP | 0.051 | 0.045 | 0.037 | 0.026 | 0.054 | 0.047 | 0.043 | 0.034 |
| HN | 0.12 | 0.08 | 0.052 | 0.041 | 0.16 | 0.098 | 0.066 | 0.49 |

Table 1: **Results on image representation and inpainting.** Reported are the MSE of the reconstructed image on test set where $N$ is the number of training samples. As can be seen, in low data regime the kernels outperform a trained hypernetwork on both tasks, and the NNGP consistently outperforms the rest.

## 5.1 Convergence of the Hyperkernel

We verified our results of Thm. 3 by constructing a simple hypernetwork, for which both $f$ and $g$ are four layered fully connected networks with ReLU activations. For the input of $f$, we used a fixed 2D vector $x = (1, -1)$. The input $z(\theta)$ of $g$ varied according to $z(\theta) = (\sin(\theta), \cos(\theta))$, where $\theta \in [-\frac{\pi}{2}, \frac{\pi}{2}]$. We then compute the empirical hyperkernel as follows, while varying the width of both $f$ and $g$:

$$\mathcal{K}^h(u, u') = \nabla_w h(x, z(\theta)) \cdot \nabla_w^\top h(x, z(\theta)) \tag{17}$$

Results are presented in Fig. 1. As can be seen, convergence to a fixed kernel is only observed in the dually wide regime, as stated in Thm. 3.

## 5.2 Training Dynamics

We consider a rotation prediction task. In this task, the hypernetwork $f$ is provided with a randomly rotated image $x$ and the primary network $g$ is provided with a rotated version $z$ of $x$ with a random angle $\alpha$. The setting is cast into a classification task, where the goal is to predict the closest value to $\alpha/360$ within $\{\alpha_i = 30i/360 \mid i = 0, \ldots, 11\}$. We experimented with the MNIST [18] and CIFAR10 [17] datasets. For each dataset we took 10000 training samples only.

We investigate the effect of the depth and width of $g$ on the training dynamics of a hypernetwork. We compared the performance of hypernetworks of various architectures to investigate the effect of the depth and width of $g$. The architectures of the hypernetwork and the primary network are as follows. The hypernetwork, $f$, is a fully-connected ReLU neural network of depth 4 and width 200. The inputs of $f$ are flattened vectors of dimension $c \cdot h^2$, where $c$ specifies the number of channels and $h$ the height/width of each image ($c = 1$ and $h = 28$ for MNIST and $c = 3$ and $h = 32$ for CIFAR10). The primary network $g$ is a fully-connected ReLU neural network of depth $\in \{3, 6, 8\}$. Since the MNIST rotations dataset is simpler, we varied the width of $g$ in $\in \{10, 50, 100\}$ and for the the CIFAR10 variation we selected the width of $g$ to be $\in \{100, 200, 300\}$. The network outputs 12 values and is trained using the cross-entropy loss.

We trained the hypernetworks for 100 epochs, using the SGD method with batch size 100 and learning rate $\mu = 0.01$. For completeness, we conducted a sensitivity study on the learning rate for both datasets, to show that the reported behaviour is consistent for any chosen learning rate, see appendix. In Fig. 2(a-c) we compare the performance of the various architectures on the MNIST and CIFAR10 rotations prediction tasks. The performance is computed as an average and standard deviation (error bars) over 100 runs. As can be seen, we observe a clear improvement in test performance as the width of $g$ increases, especially at the initialization. When comparing the three plots, we observe that when $f$ is wide and kept fixed, a deeper $g$ incurs slower training, and lower overall test performance. This is aligned with the conclusions of Thm. 1.

## 5.3 Image representation and Inpainting

We compare the performance of a hypernetwork and kernel regression with the hyperkernel on two visual tasks: functional image representation and inpainting. In the MNIST image representation task, the goal of the hypernetwork $f$ is to represent an input image via the network $g$, which receives image pixel coordinates and outputs pixel values. In the inpainting task, the goal is the same where only half of the image is observed by $f$.

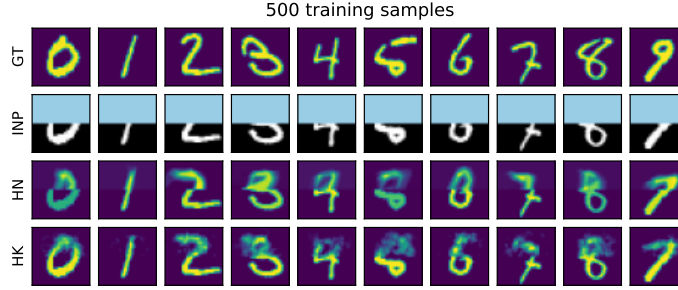

Figure 3: **Results on image inpainting. (Row 1)** ground-truth images. **(Row 2)** corresponding inputs of meta-network $f$. **(Row 3)** reconstruction by the hypernetwork. **(Row 4)** reconstruction by the hyperkernel. See Section 5.3 for experimental details.

**Problem Setup** We cast these problems as a meta-learning problem, where $f$ receives an image, and the goal of the primary network $g : [28]^2 \to [0, 1]$ is then to learn a conditional mapping from pixel coordinates to pixel values for all the pixels in the image, with the MSE as the metric. Our training dataset $S = \{(u_i, y_i)\}_{i=1}^N$ then consists of samples $u_i = (x_i, z_i)$, such that, $x_i$ are images, and $z_i$ are random pixel location (i.e., a tuple $\in [28]^2$), and $y_i$ is a label specifying the pixel value at the specified location (normalized between 0 and 1). In both experiments, training was done on randomly sampled training data of varying size, specified by $N$.

**Evaluation** We evaluate the performance of both training a hypernetwork, and using kernel regression with the hyperkernel. For kernel regression, we use the following formula to infer the pixel value of a test point $u$:

$$\left(\Theta^h(u, u_1), ..., \Theta^h(u, u_N)\right) \cdot \left(\Theta^h(U, U) + \epsilon \cdot I\right)^{-1} \cdot Y \tag{18}$$

where $\Theta^h(U, U) = (\Theta^h(u_i, u_j))_{i,j \in [N]}$ is the hyperkernel matrix evaluated on all of the training data and $Y = (y_i)_{i=1}^N$ is the vector of labels in the training dataset and $\epsilon = 0.001$.

In Tab. 1, we compare the results of the hyperkernel and the NNGP kernel with the corresponding hypernetwork. The reported numbers are averages over 20 runs. As can be seen, in the case of a small dataset, the kernels outperforms the hypernetwork, and the NNGP outperforms the rest.

## 6 Conclusions

In this paper, we apply the well-established large width analysis to hypernetwork type models. For the class of models analyzed, we have shown that a wide hypernetwork must be coupled with a wide primary network in order achieve a simplified, convex training dynamics as in standard architectures. The deeper $g$ is, the more complicated the evolution is. In the dually infinite case, when the widths of both the hyper and primary networks tend to infinity, the optimization of the hypernetwork become convex and is governed by the proposed hyperkernel. The analysis presented in this paper is limited to a specific type of hypernetworks used in the literature, typically found in the functional neural representation literature, and we leave the extension of this work to additional types of hyper models to future work.

Some of the tools developed in this study, also apply for regular NTKs. Specifically, [6] provide a conjecture, for which one of its consequences is that $\mathcal{K}_{i,j}^{(r)} = \mathcal{O}(1/n)$. In Thm. 1 we prove that this hypothesized upper bound is increasingly loose as $r$ increases, and prove an asymptotic behaviour in the order of $\mathcal{K}_{i,j}^{(r)} \sim 1/n^{r-1}$.

## Broader Impact

This work improves our understanding and design of hypernetworks and hopefully will help us improve the transparency of machine learning involving them. Beyond that, this work falls under the category of basic research and does not seem to have particular societal or ethical implications.

## Acknowledgements and Funding Disclosure

This project has received funding from the European Research Council (ERC) under the European Union's Horizon 2020 research and innovation programme (grant ERC CoG 725974). The contribution of Tomer Galanti is part of Ph.D. thesis research conducted at Tel Aviv University.

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
