[Supplementary Material]

# Appendix: On Infinite-Width Hypernetworks

**Etai Littwin**[*]
School of Computer Science
Tel Aviv University
Tel Aviv, Israel
etai.littwin@gmail.com

**Tomer Galanti**[*]
School of Computer Science
Tel Aviv University
Tel Aviv, Israel
tomerga2@tauex.tau.ac.il

**Lior Wolf**
School of Computer Science
Tel Aviv University
Tel Aviv, Israel
wolf@cs.tau.ac.il

**Greg Yang**
Microsoft Research AI
gregyang@microsoft.com

## 1 Implementation Details

### 1.1 Convergence of the Hyperkernel

In Fig. 1(a) (main text) we plot the variance of the kernel values $\mathcal{K}^h(u, u')$ in log-scale, as a function of the width of both $f$ and $g$. The variance was computed empirically over $k = 100$ normally distributed samples $w$. As can be seen, the variance of the kernel tends to zero only when both widths increase. In Fig. 1(b) (main text) we plot the value of $\mathcal{K}^h(u, u')$ and its variance for a fixed hypernetwork $f$ of width $500$ and $g$ of width $10$ or $800$. The $x$-axis specifies the value of $\theta \in \left[-\frac{\pi}{2}, \frac{\pi}{2}\right]$ and the y-axis specifies the value of the kernel. As can be seen, the expected value of the empirical kernel, $\mathcal{K}^h(u, u')$, is equal to the width-limit kernel (e.g., theoretical kernel) for both widths $10$ and $800$. In addition, convergence of the width-limit kernel is guaranteed only when the widths of both networks increase, highlighting the importance of wide architectures for both the hyper and implicit networks for stable training.

### 1.2 Image Completion and Impainting

**Architectures**  In both tasks, we used fully connected architectures, where $f$ contains two hidden layers, and $g$ contains one hidden layer. The hyperkernel used corresponds to the infinite width limit of the same architecture. For the input of $g$, we used random Fourier features [8] of the pixel coordinates as inputs for both the hyperkernel and the hypernetwork. To ease on the computational burden of computing the full kernel matrix $\Theta^h(U, U)$ when evaluating the hyperkernel, we compute smaller kernel matrices on subsets of the data $\{U_i^s\} = \{x_i^s, z_i^s\}_{s \in [10]}$, where each subset contains 1k input images $\{x_i^s\}$, and 20 random image coordinates per input, producing a kernel matrix of size $20k \times 20k$. The final output prediction is then given by:

$$\frac{1}{10} \sum_s (\Theta^h(u, u_1^s), ..., \Theta^h(u, u_N^s)) \cdot \left(\Theta^h(U^s, U^s) + \epsilon \cdot I\right)^{-1} \cdot Y^s \tag{1}$$

where $Y^s$ are the corresponding labels of the subset $U^s$. For the hypernetwork evaluation, we used the same inputs $\{U_i^s\}_{s \in [10]}$ to train the hypernetwork using a batchsize of 20, and a learning rate of 0.01 which was found to produce the best results.

---

[*]Equal Contribution

## 2 Additional Experiments

### 2.1 Sensitivity Study

To further demonstrate the behavior reported in Fig. 2 (main text), we verified that it is consistent regardless of the value of the learning rate. We used the same architectures as in the rotations prediction experiments, i.e., $f$ is a fully-connected ReLU neural network of depth 4 and width 200 and $g$ is of depth $\in \{3, 6, 8\}$ and width $\in \{50, 100, 200\}$. We vary the learning rate: $\mu = 10^{j-7}$, for $j = 0, \ldots, 7$. For each value of the learning rate, we report the average performance (and standard deviation over 100 runs) of the various architectures after 40 epochs of training.

As can be seen in Fig. 1, when $f$ is wide and kept fixed, there is a clear improvement in test performance as the width of $g$ increases, for every learning rate in which the networks provide non-trivial performance. When $f$ is wide and kept fixed, a deeper $g$ incurs slower training and lower overall test performance. We note that it might seem that the width of $g$ does not affect the performance when the learning rate is $\mu = 0.01$ in all settings except Figs. 1(c,f). Indeed, we can verify from Fig. 2 (main text) that the performance at epoch 40 is indeed similar for different widths. However, for earlier epochs, the performance improves for shallower and wider architectures.

| (a) $g$ of depth 3 | (b) $g$ of depth 6 | (c) $g$ of depth 8 |
|---|---|---|
| (d) $g$ of depth 3 | (e) $g$ of depth 6 | (f) $g$ of depth 8 |

Figure 1: Sensitivity experiment. We compare the performance of hypernetworks with an implicit network of different widths and depths after 40 epochs, when varying the learning rate. The x-axis specifies the value of the learning rate and the y-axis specifies the averaged accuracy rate at test time. (a-c) Results on MNIST and (d-f) Results on CIFAR10. In the first column, $g$'s depth is 3, in the second, it is 6 and in the third, it is 8.

### 2.2 Training wide networks with a large learning rate

Remark 1 (main text) states that one is able to train wide networks with a learning rate $\mu = o(n)$. To validate this remark, we trained shallow networks of varying width $n \in \{10^2, 10^3, 10^4, 2.5 \cdot 10^5\}$ with learning rate $\mu = \sqrt{n}$ on MNIST. As can be seen in Fig. 2, training those networks is possible despite the very large learning rate. In fact, we observe that the accuracy rate and loss improve as we increase the width of the network.

## 3 Correlation Functions

Correlation functions are products of general high order tensors representing high order derivatives of a networks output with respect to the weights. In [2] a conjecture is posed on the order of magnitude of general correlation functions involving high order derivative tensors, which arise when analysing the dynamics of gradient descent. Roughly speaking, given inputs $\{x_i\}_{i=1}^r$, the outputs of a neural

Figure 2: **Results of training wide networks with a large learning rate.** The y-axis is the **(a)** accuracy rate or **(b)** the average loss at test time. We vary the width $n \in \{10^2, 10^3, 10^4, 2.5 \cdot 10^5\}$ and take the learning rate to be $\sqrt{n}$.

network $f(x_1; w), ..., f(x_r; w) \in \mathbb{R}$ with normally distributed parameters $w \in \mathbb{R}^N$, correlation functions takes the form:

$$\sum_{\eta_{k_0}, ..., \eta_{k_r} \in [N]} \prod_{j=1}^{r} \Gamma_{\eta_{k_j+1}, ..., \eta_{k_{j+1}}}(x_j) \tag{2}$$

where

$$\Gamma_{\eta_1, ..., \eta_k}(x_j) := \frac{\partial^k f(x_j; w)}{\partial w_{\eta_1} ... \partial w_{\eta_k}} \tag{3}$$

For instance, the following are two examples of correlation functions,

$$f(x_1; w) \cdot \frac{\partial f(x_2; w)}{\partial w_{\mu_1}}, \frac{\partial^2 f(x_1; w)}{\partial w_{\mu_1} \partial w_{\mu_2}} \cdot \frac{\partial f(x_2; w)}{\partial w_{\mu_1}} \tag{4}$$

Computing the expected value of these correlation functions involve keeping track of various moments of normally distributed weights along paths, as done in recent finite width correction works [3, 5]. [2] employ the Feynman diagram to efficiently compute the expected values (order of magnitude) of general correlation functions, albeit at the cost of only being provably accurate for deep linear, or shallow ReLU networks. In this work, we analyze the asymptotic behaviour correlation functions of the form:

$$\begin{aligned} \mathcal{T}^r(x_0, ..., x_r) &:= \sum_{\eta_{k_0} ... \eta_{k_r} \in [N]} \Gamma_{\eta_{k_1}, ..., \eta_{k_r}}(x_0) \prod_{j=1}^{r} \Gamma_{\eta_{k_j}}(x_j) \\ &= \left\langle \nabla_w^{(r)} f(x_0), \bigotimes_{j=1}^{r} \nabla_w f(x_j) \right\rangle \end{aligned} \tag{5}$$

where $\nabla_w^{(r)} f(x_0)$ is a rank $r$ tensor, representing the $r$'th derivative of the output, and $\bigotimes_{j=1}^{r} \nabla_w f(x_j)$ denotes outer products of the gradients for different examples. terms of the form in Eq. 5 represent high order terms in the multivariate Taylor expansion of outputs, and are, therefore, relevant for the full understanding of training dynamics. As a consequence of Thm. 1, we prove that $\mathcal{T}^r(x_0, ..., x_r) \sim 1/n^{\max(r-1,0)}$ for vanilla neural networks, where $n$ is the width of the network. As we have shown in Sec. 3, terms of the form in Eq. 5 represent high order terms in the multivariate Taylor expansion of outputs, and are, therefore, relevant for the full understanding of training dynamics. As a consequence of Thm. 1, we prove that $\mathcal{T}^r(x_0, ..., x_r) \sim 1/n^{\max(r-1,0)}$ for vanilla neural networks, where $n$ is the width of the network.

This result is a partial solution to the open problem suggested by [2]. In their paper, they conjecture the asymptotic behaviour of general correlation functions, and predict an upper bound on the asymptotic behaviour of terms of the form in Eq. 5 in the order of $\mathcal{O}(1/n)$. Our results therefore proves a stronger version of the conjecture, while giving the exact behaviour as a function of width.

# 4  Proofs of the Main Results

**Terminology and Notations**  Throughout the appendix, we denote by $A \otimes B$ and $A \odot B$ the outer and Hadamard products of the tensors $A$ and $B$ (resp.). When considering the outer products of a sequence of tensors $\{A_i\}_{i=1}^k$, we denote, $\bigotimes_{i=1}^k A_i = A_1 \otimes \cdots \otimes A_k$. We denote by $\mathrm{sgn}(x) := x/|x|$ the sign function. The notation $X_n \sim a_n$ states that $X_n/a_n$ converges in distribution to some non-zero random variable $X$. A convenient property of this notation is that it satisfies: $X_n \cdot Y_n \sim a_n \cdot b_n$ when $X_n \sim a_n$ and $Y_n \sim b_n$. Throughout the paper, we will make use of sequential limits and denote $n_k, \ldots, n_1 \to \infty$ to express that $n_1$ tend to infinity, then $n_2$, and so on. For a given sequence of random variable $\{X_n\}_{n=1}^\infty$, we denote by $X_n \xrightarrow{d} X$ ($X_n \xrightarrow{p} X$), when $X_n$ converges in distribution (probability) to a random variable $X$.

## 4.1  Useful Lemmas

**Lemma 1.** *Let* $X_n \xrightarrow{d} X$. *Then,* $\mathrm{sgn}(X_n) \xrightarrow{d} \mathrm{sgn}(X)$.

*Proof.* We have:

$$\lim_{n\to\infty} \mathbb{P}[\mathrm{sgn}(X_n) = 1] = \lim_{n\to\infty} \mathbb{P}[X_n \geq 0] = \mathbb{P}[X \geq 0] = \mathbb{P}[\mathrm{sgn}(X) = 1] \tag{6}$$

Hence, $\mathrm{sgn}(X_n)$ converges in distribution to $\mathrm{sgn}(X)$.  □

## 4.2  Main Technical Lemma

In this section, we prove Lem. 3, which is the main technical lemma that enables us proving Thm. 1. Let $f(x; w)$ be a neural network with $H$ outputs $\{f^d(x; w)\}_{d=1}^H$. We would like to estimate the order of magnitude of the following expression:

$$\mathcal{T}_{n,i,d}^{l,i,d} := \left\langle \frac{\partial^k f^d(x_i; w)}{\partial W^{l_1} \ldots \partial W^{l_k}}, \bigotimes_{t=1}^k \frac{\partial f^{d_1}(x_{i_t}; w)}{\partial W^{l_t}} \right\rangle \tag{7}$$

where $\boldsymbol{d} = (d_1, \ldots, d_k)$, $\boldsymbol{i} = (i_1, \ldots, i_k)$ and $\boldsymbol{l} = (l_1, \ldots, l_k)$. For simplicity, when, $i_1 = \cdots = i_k = j$, we denote: $\mathcal{T}_{n,i,j,d}^{l,d} := \mathcal{T}_{n,i,d}^{l,i,d}$ and $\mathcal{T}_{n,i,j,d}^{l} := \mathcal{T}_{n,i,d}^{l,i,d}$ when $d_1 = \cdots = d_k = d$ as well.

To estimate the order of magnitude of the expression in Eq. 7, we provide an explicit expression for $\frac{\partial^k f^d(x_i; w)}{\partial W^{l_1} \ldots \partial W^{l_k}}$. First, we note that for any $w$, such that, $f^d(x_i; w)$ is $k$ times continuously differentiable at $w$, for any set $\boldsymbol{l} := \{l_1, \ldots, l_k\}$, we have:

$$\frac{\partial^k f^d(x_i; w)}{\partial W^{l_1} \ldots \partial W^{l_k}} = \frac{\partial^k f^d(x_i; w)}{\partial W^{l'_1} \ldots \partial W^{l'_k}} \tag{8}$$

where the set $\boldsymbol{l}' := \{l'_1, \ldots, l'_k\}$ is an ordered version of $\boldsymbol{l}$, i.e., the two sets consist of the same elements but $l'_1 < \cdots < l'_k$. In addition, we notice that for any multi-set $\boldsymbol{l}$, such that, $l_i = l_j$ for some $i \neq j$, then,

$$\frac{\partial^k f^d(x_i; w)}{\partial W^{l_1} \ldots \partial W^{l_k}} = 0 \tag{9}$$

since $f^d(x_i; w)$ is a neural network with a piece-wise linear activation function. Therefore, with no loss of generality, we consider $\boldsymbol{l} = \{l_1, \ldots, l_k\}$, such that, $l_1 < \cdots < l_k$. It holds that:

$$\frac{\partial^k f^d(x_i; w)}{\partial W^{l_1} \ldots \partial W^{l_k}} = \frac{1}{\sqrt{n_{l_1 - 1}}} q_{i,d}^{l_1 - 1} \otimes \mathcal{A}_{i,d}^{l_1 \to l_2} \tag{10}$$

where $\mathcal{A}_{i,d}^{l_1 \to l_2}$ is a $2k - 1$ tensor, defined as follows:

$$\mathcal{A}_{i,d}^{l_j \to l_{j+1}} = \begin{cases} \frac{1}{\sqrt{n_{l_{j+1} - 1}}} C_{i,d}^{l_j \to l_{j+1}} \otimes \mathcal{A}_{i,d}^{l_{j+1} \to l_{j+2}} & 1 < j < k - 1 \\ \frac{1}{\sqrt{n_{l_k - 1}}} C_{i,d}^{l_{k-1} \to l_k} \otimes C_{i,d}^{l_k \to L} & j = k - 1 \end{cases} \tag{11}$$

where:

$$C_{i,d}^{l_j \to l_{j+1}} = \begin{cases} \sqrt{2} Z_{i,d}^{l_{j+1}-1} P_{i,d}^{l_j \to l_{j+1}-1} & l_{j+1} \neq L \\ P_{i,d}^{l_j \to L} & else \end{cases} \tag{12}$$

and:

$$P_i^{u \to v} = \prod_{l=u}^{v-1} \left( \sqrt{\frac{2}{n_l}} W^{l+1} Z_i^l \right) \text{ and } Z_i^l = \text{diag}(\dot{\sigma}(y^l(x_i))) \tag{13}$$

The individual gradients can be expressed using:

$$\frac{\partial f_w^{d_j}(x_{i_j})}{\partial W^{l_j}} = \frac{q_{i_j,d_j}^{l_j-1} \otimes C_{i_j,d_j}^{l_j \to L}}{\sqrt{n_{l_j-1}}} \tag{14}$$

Note that the following holds for any $u < v < h \leq L$:

$$C_{i,d}^{u \to h} = C^{v \to h} \frac{W^v}{\sqrt{n_{v-1}}} C_{i,d}^{u \to v} \text{ and } C_{i,d}^{u \to L} = C_{i,d}^{v-1 \to L} P_{i,d}^{u \to v-1} \tag{15}$$

In the following, given the sets $\boldsymbol{l} = \{l_1, \ldots, l_k\}$, $\boldsymbol{i} = \{i_1, \ldots, i_k\}$ and $\boldsymbol{d} = \{d_1, \ldots, d_k\}$, we derive the limit of $\mathcal{T}_{n,i,d}^{\boldsymbol{l,i,d}}$ using elementary tensor algebra. By Eqs. 14 and 10, we see that:

$$\begin{aligned}
\mathcal{T}_{n,i,d}^{\boldsymbol{l,i,d}} &= \left\langle \bigotimes_{t=1}^{k} \frac{\partial f^{d_t}(x_{i_t}; w)}{\partial W^{l_t}}, \frac{q_{i,d}^{l_1-1}}{\sqrt{n_{l_1-1}}} \otimes \frac{C_{i,d}^{l_1 \to l_2}}{\sqrt{n_{l_2-1}}} \otimes \ldots \otimes \frac{C_{i,d}^{l_{r-1} \to l_k}}{\sqrt{n_{l_k-1}}} \otimes C_{i,d}^{l_k \to L} \right\rangle \\
&= \frac{1}{n_{l_1-1}} \left\langle q_{i,d}^{l_1-1}, q_{i_1,d_1}^{l_1-1} \right\rangle \cdot \left\langle C_{i_k,d_k}^{l_k \to L}, C_{i,d}^{l_k \to L} \right\rangle \prod_{j=1}^{k-1} \left\langle \frac{C_{i_j,d_j}^{l_j \to L} \otimes q_{i_{j+1},d_{j+1}}^{l_{j+1}-1}}{n_{l_{j+1}-1}}, C_{i,d}^{l_j \to l_{j+1}} \right\rangle
\end{aligned} \tag{16}$$

We recall the analysis of [9] showing that in the infinite width limit, with $n = \min(n_1 \ldots, n_{L-1}) \to \infty$, every pre-activation $y^l(x)$ of $f(x; w)$ at hidden layer $l \in [L]$ has all its coordinates tending to i.i.d. centered Gaussian processes of covariance $\Sigma^l(x, x') : \mathbb{R}^{n_0} \times \mathbb{R}^{n_0} \to \mathbb{R}$ defined recursively as follows:

$$\begin{aligned}
\Sigma^0(x, x') &= x^\top x', \\
\Lambda^l(x, x') &= \begin{bmatrix} \Sigma^{l-1}(x, x) & \Sigma^{l-1}(x, x') \\ \Sigma^{l-1}(x', x) & \Sigma^{l-1}(x', x') \end{bmatrix} \in \mathbb{R}^{2 \times 2}, \\
\Sigma^l(x, x') &= \mathbb{E}_{(u,v) \sim \mathcal{N}(0, \Lambda^{l-1}(x,x'))}[\sigma(u)\sigma(v)]
\end{aligned} \tag{17}$$

In addition, we define the derivative covariance as follows:

$$\dot{\Sigma}^l(x, x') = \mathbb{E}_{(u,v) \sim \mathcal{N}(0, \Lambda^{l-1}(x,x'))}[\dot{\sigma}(u)\dot{\sigma}(v)] \tag{18}$$

when considering $x = x_i$ and $x' = x_j$ from the training set, we simply write $\Sigma_{i,j}^l := \Sigma^l(x_i, x_j)$ and $\dot{\Sigma}_{i,j}^l = \dot{\Sigma}^l(x_i, x_j)$.

**Lemma 2.** *The following holds:*

1. *For $n_{v-1}, \ldots, n_1 \to \infty$, we have: $P_i^{u \to v}(P_j^{u \to v})^\top \xrightarrow{d} \prod_{l=u}^{v-1} \dot{\Sigma}_{i,j}^l I$.*

2. *For $n_{L-1}, \ldots, n_1 \to \infty$, we have: $P_{i,d_1}^{u \to L}(P_{j,d_2}^{u \to L})^\top \xrightarrow{d} \prod_{l=u}^{L-1} \dot{\Sigma}_{i,j}^l \delta_{d_1=d_2}$.*

3. *For $n_v, \ldots, n_1 \to \infty$, we have: $\frac{(q_i^v)^\top q_j^v}{n_v} \xrightarrow{d} \Sigma_{i,j}^v$.*

*Here, $\delta_T$ is an indicator that returns 1 if $T$ is true and 0 otherwise.*

*Proof.* See [1].

**Lemma 3.** *Let $k \geq 0$ and sets $\boldsymbol{l} = \{l_1, \ldots, l_k\}$, $\boldsymbol{i} = \{i_1, \ldots, i_k\}$ and $\boldsymbol{d} = \{d_1, \ldots, d_k\}$. We have:*

$$n^{\max(k-1,0)} \cdot \mathcal{T}_{n,i,d}^{\boldsymbol{l,i,d}} \xrightarrow{d} \begin{cases} \delta_{\boldsymbol{d}} \cdot \prod_{j=1}^{k-1} \mathcal{G}_j & k > 1 \\ const & k = 1 \end{cases} \tag{19}$$

*as $n \to \infty$. Here, $\mathcal{G}_1, \ldots, \mathcal{G}_{k-1}$ are centered Gaussian variables with finite, non-zero variances, and $\delta_{\boldsymbol{d}} := \delta(d_1 = \ldots = d_k = d)$.*

*Proof.* The case $k = 0$ is trivial. Let $k \geq 1$. By Eq. 16, it holds that:

$$
\begin{aligned}
&n^{k-1}\mathcal{T}_{n,i,d}^{l,i,d} \\
=&n^{k-1}\frac{\left\langle q_{i,d}^{l_1-1}, q_{i_1,d_1}^{l_1-1}\right\rangle\left\langle C_{i_k,d_k}^{l_k\to L}, C_{i,d}^{l_k\to L}\right\rangle}{n} \cdot \prod_{j=1}^{k-1}\left\langle \frac{C_{i_j,d_j}^{l_j\to L} \otimes q_{i_{j+1},d_{j+1}}^{l_{j+1}-1}}{n}, C_{i,d}^{l_j\to l_{j+1}}\right\rangle \\
=&\frac{\left\langle q_{i,d}^{l_1-1}, q_{i_1,d_1}^{l_1-1}\right\rangle\left\langle C_{i_k,d_k}^{l_k\to L}, C_{i,d}^{l_k\to L}\right\rangle}{n} \cdot \prod_{j=1}^{k-1}\left\langle C_{i_j,d_j}^{l_j\to L} \otimes q_{i_{j+1},d_{j+1}}^{l_{j+1}-1}, C_{i,d}^{l_j\to l_{j+1}}\right\rangle
\end{aligned}
\tag{20}
$$

Note that intermediate activations do not depend on the index $d_j$, and so we remove the dependency on $d_j$ in the relevant terms. Next, by applying Lem. 2,

$$
\frac{\left\langle q_i^{l_1-1}, q_{i_1}^{l_1-1}\right\rangle\left\langle C_{i_k,d_k}^{l_k\to L}, C_{i,d}^{l_k\to L}\right\rangle}{n} \xrightarrow{d} \Sigma_{i,i_1}^{l_1-1}\left(\prod_{j=l_k}^{L}\dot{\Sigma}_{i,i_k}^{l_j}\right)\delta_{\boldsymbol{d}}
\tag{21}
$$

Expanding the second term using Eq. 15:

$$
\begin{aligned}
&\left\langle C_{i_j,d_j}^{l_j\to L} \otimes q_{i_{j+1}}^{l_{j+1}-1}, C_{i,d}^{l_j\to i_{j+1}}\right\rangle \\
=&\, C_{i_j,d_j}^{l_j\to L}C_i^{l_j\to i_{j+1}}q_{i_{j+1}}^{l_{j+1}-1} \\
=&\, C_{i_j,d_j}^{l_{j+1}-1\to L}P_{i_j}^{l_j\to l_{j+1}-1}(P_i^{l_j\to l_{j+1}-1})^\top \sqrt{2} \cdot Z_i^{l_{j+1}-1}q_{i_{j+1}}^{l_{j+1}-1} \\
=&\, \sqrt{2} \cdot \left\langle C_{i_j,d_j}^{l_{j+1}-1\to L} \otimes (Z_i^{l_{j+1}-1}q_{i_{j+1}}^{l_{j+1}-1}), P_{i_j}^{l_j\to l_{j+1}-1}(P_i^{l_j\to l_{j+1}-1})^\top\right\rangle \\
=&\, \sqrt{2} \cdot C_{i_j,d_j}^{l_{j+1}-1\to L}P_{i_j}^{l_j\to l_{j+1}-1}(P_i^{l_j\to l_{j+1}-1})^\top Z_i^{l_{j+1}-1}q_{i_{j+1}}^{l_{j+1}-1}
\end{aligned}
\tag{22}
$$

Since the limit of a product equals the product of limits (when the limits exist), it holds that (after taking the limit of the right term in the above inner product):

$$
P_{i_j}^{l_j\to l_{j+1}-1}(P_i^{l_j\to l_{j+1}-1})^\top \xrightarrow{d} \prod_{l=l_j}^{l_{j+1}-2}\dot{\Sigma}_{i,i_j}^{l}
\tag{23}
$$

Recall that in the infinite width limit, when conditioned on the outputs $q_i^{l-1}, q_j^{l-1}$ the pre activations $y_i^l, y_j^l$ are GPs. Hence, when conditioned on the outputs $q_i^{l-1}, q_j^{l-1}$, the diagonal components of the product $Z_i^l Z_j^l$ are independent. The GP behaviour argument then applies to terms $C_{i_j,d_j}^{l_{j+1}-1\to L}Z_i^{l_{j+1}-1}q_{i_{j+1}}^{l_{j+1}-1}$. Assigning:

$$
\xi_j = C_{i_j,d_j}^{l_{j+1}-1\to L}Z_i^{l_{j+1}-1}q_{i_{j+1}}^{l_{j+1}-1}
\tag{24}
$$

and their limits:

$$
\xi_j \xrightarrow{d} \mathcal{G}_j
\tag{25}
$$

and denoting $\boldsymbol{\xi} = [\xi_1, ..., \xi_{k-1}]$, and $\boldsymbol{\mathcal{G}} = [\mathcal{G}_1, ..., \mathcal{G}_{k-1}]$, it holds using the multivariate Central Limit theorem:

$$
\boldsymbol{\xi} \xrightarrow{d} \boldsymbol{\mathcal{G}}
\tag{26}
$$

Using the Mann-Wald theorem [6] (where we take the mapping as the product pooling of $\boldsymbol{\xi}$), we have that:

$$
\prod_{j=1}^{k-1}\xi_j \xrightarrow{d} \prod_{j=1}^{k-1}\mathcal{G}_j
\tag{27}
$$

Finally, by Slutsky's theorem,

$$
n^{k-1}\mathcal{T}_{n,i,d}^{l,i,d} \xrightarrow{d} \Sigma_{i,i_1}^{l_1-1}\left(\prod_{j=l_k}^{L}\dot{\Sigma}_{i,i_k}^{l_j}\right)\prod_{j=1}^{k-1}\left(\left[\prod_{l=l_j}^{l_{j+1}-2}\dot{\Sigma}_{i,i_j}^{l}\right]\cdot\sqrt{2}\cdot\mathcal{G}_j\right)\cdot\delta_{\boldsymbol{d}}
\tag{28}
$$

$\square$

### 4.3 Proof of Thm. 1

Since we assume that $g$ is a finite neural network, i.e., $m_l < \infty$ for all $l \in [H]$, throughout the proofs with no loss of generality we assume that $m_1 = \cdots = m_H = 1$.

**Lemma 4.** *Let $h(u; w) = g(z; f(x; w))$ be a hypernetwork. We have:*

$$\mathcal{K}_{i,j}^{(r)} = \sum_{\substack{\alpha_1 + \cdots + \alpha_H = r \\ \alpha_1, \ldots, \alpha_H \geq 0}} \frac{r!}{\alpha_1! \cdots \alpha_H!} \cdot z_i \cdot \left[ \prod_{j=1}^{H-1} \dot{\phi}(g_i^j) \right] \cdot \prod_{d=1}^{H} \left\langle \nabla_w^{(\alpha_d)} f_i^d, (\nabla_w h_j)^{\alpha_d} \right\rangle \tag{29}$$

*Proof.* By the higher order product rule and the fact that the second derivative of a piece-wise linear function is $0$ everywhere:

$$\nabla_w^{(r)} h_i = \sum_{\substack{\alpha_1 + \cdots + \alpha_H = r \\ \alpha_1, \ldots, \alpha_H \geq 0}} \frac{r!}{\alpha_1! \cdots \alpha_H!} \cdot z_i \cdot \nabla_w^{(\alpha_H)} f_i^H \bigotimes_{d=1}^{H-1} D_{H-d} \tag{30}$$

where

$$D_d := \dot{\phi}(g_i^d) \cdot \nabla_w^{(\alpha_d)} f_i^d \tag{31}$$

In addition, by elementary tensor algebra, we have:

$$\begin{aligned}
\mathcal{K}_{i,j}^{(r)} &= \langle \nabla_w^{(r)} h_i, (\nabla_w h_j)^r \rangle \\
&= \sum_{\substack{\alpha_1 + \cdots + \alpha_H = r \\ \alpha_1, \ldots, \alpha_H \geq 0}} \frac{r!}{\alpha_1! \cdots \alpha_H!} z_i \cdot \left\langle \nabla_w^{(\alpha_H)} f_i^H \cdot \bigotimes_{d=1}^{H-1} D_{H-d}, (\nabla_w h_j)^r \right\rangle \\
&= \sum_{\substack{\alpha_1 + \cdots + \alpha_H = r \\ \alpha_1, \ldots, \alpha_H \geq 0}} \frac{r!}{\alpha_1! \cdots \alpha_H!} z_i \cdot \left\langle \nabla_w^{(\alpha_H)} f_i^H, (\nabla_w h_j)^{\alpha_H} \right\rangle \\
&\qquad \cdot \prod_{d=1}^{H-1} \left\langle \dot{\phi}(g_i^{H-d}) \cdot \nabla_w^{(\alpha_{H-d})} f_i^{H-d}, (\nabla_w h_j)^{\alpha_{H-d}} \right\rangle \\
&= \sum_{\substack{\alpha_1 + \cdots + \alpha_H = r \\ \alpha_1, \ldots, \alpha_H \geq 0}} \frac{r!}{\alpha_1! \cdots \alpha_H!} \cdot z_i \cdot \left[ \prod_{d=1}^{H-1} \dot{\phi}(g_i^d) \right] \cdot \prod_{d=1}^{H} \left\langle \nabla_w^{(\alpha_d)} f_i^d, (\nabla_w h_j)^{\alpha_d} \right\rangle
\end{aligned} \tag{32}$$

$\square$

**Lemma 5.** *Let $h(u; w) = g(z; f(x; w))$ be a hypernetwork. In addition, let,*

$$\forall d \in [H]: \quad h_j^d := a_j^{d-1} \prod_{t=1}^{H-d} f_j^{H-t+1} \cdot \dot{\phi}(g_j^{H-t}) \tag{33}$$

*We have:*

$$\left\langle \nabla^{(\alpha_d)} f_i^d, (\nabla_w h_j)^{\alpha_d} \right\rangle = \sum_{\boldsymbol{l} \in [L]^{\alpha_d}} \sum_{\boldsymbol{d} \in [H]^{\alpha_d}} \left( \prod_{k=1}^{\alpha_d} h_j^{d_k} \right) \cdot \mathcal{T}_{n,i,j,d}^{\boldsymbol{l},\boldsymbol{d}} \tag{34}$$

*Proof.* We have:

$$\left\langle \nabla^{(\alpha_d)} f_i^d, (\nabla_w h_j)^{\alpha_d} \right\rangle = \sum_{\boldsymbol{l} \in [L]^{\alpha_d}} \left\langle \frac{\partial^{\alpha_d} f_i^d}{\partial W^{l_1} \ldots \partial W^{l_{\alpha_d}}}, \bigotimes_{k=1}^{\alpha_d} \frac{\partial h_j}{\partial W^{l_k}} \right\rangle \tag{35}$$

By the product rule:

$$\frac{\partial h_j}{\partial W^{l_k}} = \sum_{d=1}^{H} \left[ \prod_{t=1}^{H-d} f_j^{H-t+1} \cdot \dot{\phi}(g_j^{H-t}) \right] \cdot \frac{\partial f_j^d}{\partial W^{l_k}} \cdot a_j^{d-1} = \sum_{d=1}^{H} h_j^d \cdot \frac{\partial f_j^d}{\partial W^{l_k}} \tag{36}$$

Hence,

$$\bigotimes_{k=1}^{\alpha_d} \frac{\partial h_j}{\partial W^{l_k}} = \sum_{\boldsymbol{d} \in [H]^{\alpha_d}} \left( \prod_{k=1}^{\alpha_d} h_j^{d_k} \right) \bigotimes_{k=1}^{\alpha_d} \frac{\partial f_j^{d_k}}{\partial W^{l_k}} \tag{37}$$

In particular,

$$\left\langle \nabla^{(\alpha_d)} f_i^d, (\nabla_w h_j)^{\alpha_d} \right\rangle = \sum_{\boldsymbol{l} \in [L]^{\alpha_d}} \sum_{\boldsymbol{d} \in [H]^{\alpha_d}} \left( \prod_{k=1}^{\alpha_d} h_j^{d_k} \right) \cdot \mathcal{T}_{n,i,j,d}^{\boldsymbol{l},\boldsymbol{d}} \tag{38}$$

$\square$

**Theorem 1** (Higher order terms for hypernetworks). *Let $h(u) = g(z; f(x))$ for a hypernetwork $f$ and an implicit network $g$. Then, we have:*

$$\mathcal{K}_{i,j}^{(r)} \sim \begin{cases} n^{H-r} & \text{if } r > H \\ 1 & \text{otherwise.} \end{cases} \tag{39}$$

*Proof.* Throughout the proof, in order to derive certain limits of various sequences of random variables, we implicitly make use of the Mann-Wald theorem [6]. For simplicity, oftentimes, we will avoid explicitly stating when this theorem is applied. As a general note, the repeated argument is as follows: terms, such as, $n^{\max(\alpha_d - 1, 0)} \cdot \mathcal{T}_{n,i,j,d}^{\boldsymbol{l},\boldsymbol{d}}$, $\mathcal{Q}_{n,j}^{\boldsymbol{d}}$, $g_i^d$, etc', (see below) can be expressed as continuous mappings of jointly convergent random variables. Hence, they jointly converge, and continuous mappings over them converge as well.

By Lems. 4 and 5, we have:

$$\mathcal{K}_{i,j}^{(r)} = \sum_{\substack{\alpha_1 + \cdots + \alpha_H = r \\ \alpha_1, \ldots, \alpha_H \geq 0}} \frac{r!}{\alpha_1! \cdots \alpha_H!} \cdot z_i \cdot \left[ \prod_{d=1}^{H-1} \dot{\phi}(g_i^d) \right] \cdot \prod_{d=1}^{H} \sum_{\boldsymbol{l} \in [H]^{\alpha_d}} \sum_{\boldsymbol{d} \in [H]^{\alpha_d}} \mathcal{Q}_{n,j}^{\boldsymbol{d}} \cdot \mathcal{T}_{n,i,j,d}^{\boldsymbol{l},\boldsymbol{d}} \tag{40}$$

where $\mathcal{Q}_{n,j}^{\boldsymbol{d}} := \left( \prod_{k=1}^{\alpha_d} h_j^{d_k} \right)$. By the Mann-Wald theorem [6], $g_i^d$ converges to some random variable $\mathcal{U}_i^d$. If $\dot{\phi}$ is a continuous function, then $\dot{\phi}(g_i^d)$ converges to $\dot{\phi}(\mathcal{U}_i^d)$. If $\phi$ is the ReLU activation function, by Lem. 1, $\dot{\phi}(g_i^d) = \text{sgn}(g_i^d)$ converges to $\text{sgn}(\mathcal{U}_i^d)$ in distribution. We notice that $\mathcal{Q}_{n,j}^{\boldsymbol{d}}$ converges in distribution to some random variable $\mathcal{Q}_j^{\boldsymbol{d}}$.

The proof is divided into two cases: $H = 1$ and $H > 1$.

**Case $H = 1$:** First, we note that for $H = 1$ and $d \in [H]$ (i.e., $d = 1$), we have:

$$h_j^d = a_j^{d-1} \cdot \prod_{t=1}^{H-d} f_j^{H-t+1} \cdot \dot{\sigma}(g_j^{H-t}) = a_j^0 = z_j \tag{41}$$

In addition, $\prod_{d=1}^{H-1} \dot{\sigma}(g_i^d) = 1$ as it is an empty product. Therefore, we can rewrite:

$$\mathcal{K}_{i,j}^{(r)} = z_i \cdot z_j^r \sum_{\boldsymbol{l} \in [H]^r} \sum_{\boldsymbol{d} \in [H]^r} \mathcal{T}_{n,i,j,d}^{\boldsymbol{l},\boldsymbol{d}} \tag{42}$$

By Lem. 3, for $r = 1$, the above tends to a constant as $n \to \infty$. For $r > 1$, $n^{r-1} \cdot \mathcal{T}_{n,i,j,d}^{\boldsymbol{l},\boldsymbol{d}}$ converges in distribution to zero for all $\boldsymbol{d} \neq (d, \ldots, d)$ and converges to a non-constant random variable $\mathcal{T}_{i,j,d}^{\boldsymbol{l}}$ otherwise. Hence, by the Mann-Wald theorem [6],

$$n^{r-1} \cdot \mathcal{K}_{i,j}^{(r)} \xrightarrow{d} z_i \cdot z_j^r \sum_{\boldsymbol{l} \in [H]^r} \mathcal{T}_{i,j,d}^{\boldsymbol{l}} \tag{43}$$

which is a non-zero random variable.

**Case $H > 1$:** By Lem. 3, $n^{\alpha_d - 1} \cdot \mathcal{T}_{n,i,j,d}^{l,d}$ converges in distribution to zero for all $d \neq (d, \dots, d)$. Therefore, in these cases, by Slutsky's theorem, $n^{\alpha_d - 1} \cdot \mathcal{Q}_{n,j}^d \cdot \mathcal{T}_{n,i,j,d}^{l,d}$ converges to zero in distribution. On the other hand, for each $l \in [H]^{\alpha_d}$, $d \in [H]$ and $d = (d, \dots, d)$, by Lem. 3, we have:

$$n^{\alpha_d - 1} \cdot \mathcal{Q}_{n,j}^d \cdot \mathcal{T}_{n,i,j,d}^l \xrightarrow{d} \mathcal{Q}_j^d \cdot \mathcal{T}_{i,j,d}^l \tag{44}$$

In particular,

$$n^{\max(\alpha_d - 1, 0)} \sum_{l \in [H]^{\alpha_d}} \sum_{d \in [H]^{\alpha_d}} \cdot \mathcal{Q}_{n,j}^d \cdot \mathcal{T}_{n,i,j,d}^l \xrightarrow{d} \sum_{l \in [H]^{\alpha_d}} \sum_{d \in [H]} \mathcal{Q}_j^d \cdot \mathcal{T}_{i,j,d}^l \tag{45}$$

Consider the case where $r \geq H$. In this case, for any $\alpha_1, \dots, \alpha_H$, such that, there are $t > 1$ indices $i \in [H]$, such that, $\alpha_i = 0$. The following random variable converges in distribution:

$$X_n := n^{r - (H - t)} \cdot \prod_{d=1}^{H} \sum_{l \in [H]^{\alpha_d}} \sum_{d \in [H]^{\alpha_d}} \mathcal{Q}_{n,j}^d \cdot \mathcal{T}_{n,i,j,d}^{l,d} \tag{46}$$

Therefore, by Slutsky's theorem:

$$n^{r - H} \cdot \prod_{d=1}^{H} \sum_{l \in [H]^{\alpha_d}} \sum_{d \in [H]^{\alpha_d}} \mathcal{Q}_{n,j}^d \cdot \mathcal{T}_{n,i,j,d}^{l,d} = n^{-t} \cdot X_n \xrightarrow{d} 0 \tag{47}$$

We have:

$$n^{r-H} \cdot \left\langle \nabla_w^{(r)} h_i, (\nabla_w h_j)^r \right\rangle$$

$$= n^{r-H} \sum_{\substack{\alpha_1 + \dots + \alpha_H = r \\ \alpha_1, \dots, \alpha_H \geq 0}} \frac{r!}{\alpha_1! \cdots \alpha_H!} \cdot z_i \cdot \left[ \prod_{d=1}^{H-1} \dot{\sigma}(g_i^d) \right] \cdot \prod_{d=1}^{H} \sum_{l \in [H]^{\alpha_d}} \sum_{d \in [H]^{\alpha_d}} \left( \prod_{k=1}^{\alpha_d} h_j^{d_k} \right) \mathcal{T}_{n,i,j,d}^{l,d}$$

$$= \sum_{\substack{\alpha_1 + \dots + \alpha_H = r \\ \alpha_1, \dots, \alpha_H \geq 0}} \frac{r!}{\alpha_1! \cdots \alpha_H!} \cdot z_i \cdot \left[ \prod_{d=1}^{H-1} \dot{\sigma}(g_i^d) \right] \cdot \prod_{d=1}^{H} n^{\alpha_d - 1} \sum_{l \in [H]^{\alpha_d}} \sum_{d \in [H]^{\alpha_d}} \mathcal{Q}_{n,j}^d \cdot \mathcal{T}_{n,i,j,d}^{l,d}$$

$$\xrightarrow{d} \sum_{\substack{\alpha_1 + \dots + \alpha_H = r \\ \alpha_1, \dots, \alpha_H \geq 1}} \frac{r!}{\alpha_1! \cdots \alpha_H!} \cdot z_i \cdot \left[ \prod_{d=1}^{H-1} \operatorname{sgn}(\mathcal{U}_i^d) \right] \cdot \prod_{d=1}^{H} \sum_{l \in [H]^{\alpha_d}} \mathcal{Q}_j^d \cdot \mathcal{T}_{i,j,d}^l$$
$$\tag{48}$$

which is a non-constant random variable.

Next, we consider the case when $r \leq H$. By Lem. 3, for any $\alpha_d \geq 2$, the term $\mathcal{T}_{n,i,j,d}^{l,d}$ tends to zero as $n \to \infty$. In addition, $\mathcal{Q}_{n,j}^d$ converges in distribution. Therefore, for any $\alpha_d \geq 2$, we have:

$$\sum_{l \in [L]^{\alpha_d}} \sum_{d \in [H]^{\alpha_d}} \mathcal{Q}_{n,j}^d \cdot \mathcal{T}_{n,i,j,d}^{l,d} \xrightarrow{d} 0 \tag{49}$$

Hence, for any $\alpha_1, \dots, \alpha_H \geq 0$, such that, there is at least one $\alpha_d \geq 2$, we have:

$$\prod_{d=1}^{H} \sum_{l \in [L]^{\alpha_d}} \sum_{d \in [H]^{\alpha_d}} \mathcal{Q}_{n,j}^d \cdot \mathcal{T}_{n,i,j,d}^{l,d} \xrightarrow{d} 0 \tag{50}$$

On the other hand, for any $0 \leq \alpha_1, \dots, \alpha_H \leq 1$, the terms $\{\mathcal{T}_{n,i,d}^{l,i,d}\}$, $\{g_i^d\}$ and $\{\mathcal{Q}_{n,j}^d\}$ converge jointly in distribution to some random variables $\{\mathcal{T}_{i,d}^{l,i,d}\}$, $\{\operatorname{sgn}(\mathcal{U}_i^d)\}$ and $\{\mathcal{Q}_j^d\}$ as $n \to \infty$. Hence,

$$\left\langle \nabla_w^{(r)} h_i, (\nabla_w h_j)^r \right\rangle \xrightarrow{d} \sum_{\substack{\alpha_1 + \dots + \alpha_H = r \\ 0 \leq \alpha_1, \dots, \alpha_H \leq 1}} r! \cdot \left[ \prod_{d=1}^{H-1} \operatorname{sgn}(\mathcal{U}_i^d) \right] \cdot \prod_{d=1}^{H} \sum_{l \in [L]^{\alpha_d}} \sum_{d \in [H]^{\alpha_d}} \mathcal{Q}_j^d \cdot \mathcal{T}_{i,j,d}^{l,d} \tag{51}$$

which is a non-constant random variable. $\qquad\square$

### 4.4 Proofs of the Results in Sec. 4

**Theorem 2** (Hypernetworks as GPs). *Let $h(u) = g(z; f(x))$ be a hypernetwork. For any pair of inputs $u = (x, z)$ and $u' = (x', z')$, let $\Sigma^0(z, z') = \frac{z^\top z'}{m_0}$, $S^0(x, x') = \frac{x^\top x'}{n_0}$. Then, it holds for any unit $i$ in layer $0 < l \leq H$ of the implicit network:*

$$g_i^l(z; f(x)) \xrightarrow{d} \mathcal{G}_i^l(u) \tag{52}$$

*as $m, n \to \infty$ sequentially. Here, $\{\mathcal{G}_i^l(u)\}_{i=1}^{m_l}$ are independent Gaussian processes, such that, $(\mathcal{G}_i^l(u), \mathcal{G}_i^l(u')) \sim \mathcal{N}(0, \Lambda^l(u, u'))$ defined by the following recursion:*

$$\Lambda^{l+1}(u, u') = \begin{pmatrix} \Sigma^l(u, u) & \Sigma^l(u', u) \\ \Sigma^l(u, u') & \Sigma^l(u', u') \end{pmatrix} \odot \begin{pmatrix} S^L(x, x) & S^L(x', x) \\ S^L(x, x') & S^L(x', x') \end{pmatrix} \tag{53}$$

$$\Sigma^l(u, u') = 2 \mathop{\mathbb{E}}_{(u,v) \sim \mathcal{N}(0, \Lambda^l)} [\sigma(u) \cdot \sigma(v)] \tag{54}$$

*where $S^L(x, x')$ is defined recursively:*

$$S^l(x, x') = 2 \mathop{\mathbb{E}}_{(u,v) \sim \mathcal{N}(0, \Gamma^l)} [\sigma(u) \cdot \sigma(v)] \text{ and } \Gamma^l(x, x') = \begin{pmatrix} S^l(x, x) & S^l(x', x) \\ S^l(x, x') & S^l(x', x') \end{pmatrix} \tag{55}$$

*Proof.* By [9], taking the width $n = \min(n_1, ..., n_{L-1})$ to infinity, the outputs $V^d(x; w) := f^d(x; w)$ are governed by a centered Gaussian process, such that, the entries $V_{i,j}^d(x; w)$, given some input $x$, are independent and identically distributed. Moreover, it holds that:

$$\left( V_{i,j}^d(x; w), V_{i,j}^d(x'; w) \right) \sim \mathcal{N}\left( 0, S^L(x, x') \right). \tag{56}$$

with $S^L(x, x')$ as defined in Eq. 54. For the function $h(u; w) = g(z; f(x; w))$, it holds for the first layer:

$$g^1(z; f(x; w)) = \sqrt{\frac{1}{m_0}} V^1(x; w) z \tag{57}$$

After taking the limit $n = \min(n_1, ..., n_{L-1})$ to infinity, the implicit network $g$ is fed with Gaussian distributed weights. And so $g^1(z; f(x; w))$ also converges to a Gaussian process, such that:

$$(g^1(z; f(x; w))_i, g^1(z'; f(x'; w))_i) \sim \mathcal{N}(0, \Lambda^1) \tag{58}$$

where:

$$\Lambda^1 = \frac{1}{m_0} \begin{pmatrix} S^L(x, x) z^\top z & S^L(x', x) z'^\top z \\ S^L(x, x') z^\top z' & S^L(x', x') z'^\top z' \end{pmatrix} \tag{59}$$

In a similar fashion to the standard feed forward case, the pre-activations $g^l(z; f(x; w))$ converge to Gaussian processes as we let $m = \min(m_1, ..., m_{H-1})$ tend to infinity, with a covariance defined recursively:

$$\Sigma^l(u, u') = \sqrt{2} \mathop{\mathbb{E}}_{(u,v) \sim \mathcal{N}(0, \Lambda^l)} [\sigma(u) \sigma(v)] \tag{60}$$

where,

$$\Lambda^l = \begin{pmatrix} S^L(x, x) \cdot \Sigma^{l-1}(u, u) & S^L(x', x) \cdot \Sigma^{l-1}(u', u) \\ S^L(x, x') \cdot \Sigma^{l-1}(u, u') & S^L(x', x') \cdot \Sigma^{l-1}(u', u') \end{pmatrix} \tag{61}$$

and

$$\Sigma^0(z, z') = \frac{1}{m_0} z^\top z' \tag{62}$$

proving the claim. $\square$

**Corollary 1.** *Let $h(u) = g(z; f(x))$ be a hypernetwork. For any $0 < l \leq H$, there exists a function $\mathcal{F}^l$, such that, for all pairs of inputs $u = (x, z)$ and $u' = (x', z')$, it holds that:*

$$\Lambda^H(u, u') = \mathcal{F}\left( \Sigma^0(z, z'), S^0(x, x') \right) \tag{63}$$

*Proof.* We prove that $\Lambda^l(u, u')$ is a function of $S^0(x, x')$ and $\Sigma^0(u, u')$ by induction. First, we note that $\Lambda^1(u, u')$ is a function of $S^L(x, x')$ and $\Sigma^0(u, u')$ by definition. By the recursive definition of $S^L(x, x')$, it is a function of $S^0(x, x')$. Therefore, $\Lambda^1(u, u')$ can be simply represented as a function of $S^0(x, x')$ and $\Sigma^0(u, u')$. We assume by induction that $\Lambda^l(u, u')$ is a function of $S^0(x, x')$ and $\Sigma^0(u, u')$. We would like to show that $\Lambda^{l+1}(u, u')$ is a function of $S^0(x, x')$ and $\Sigma^0(u, u')$. By definition, $\Lambda^{l+1}(u, u')$ is a function of $S^L(x, x')$ and $\Sigma^l(u, u')$. In addition, $\Sigma^l(u, u')$ is a function of $\Lambda^l(u, u')$. Hence, by induction, $\Sigma^l(u, u')$ is simply a function of $S^0(x, x')$ and $\Sigma^0(u, u')$. Since $S^L(x, x')$ is a function of $S^0(x, x')$, we conclude that one can represent $\Lambda^{l+1}(u, u')$ as a function of $S^0(x, x')$ and $\Sigma^0(u, u')$. $\qquad\square$

**Remark 1.** *Let $p(z) = [\cos(W_i^1 z + b_i^1)]_{i=1}^k$ be a Fourier features preprocessing, where $W_{i,j}^1 \sim \mathcal{N}(0, 1)$ and biases $b_i \sim U[-\pi, \pi]$. Let $h(u) = g(p(z); f(x))$ be a hypernetwork, with $z$ preprocessed according to $p$. Let $u = (x, z)$ and $u' = (x', z')$ be two pairs of inputs. Then, $\Lambda^l(u, u')$ is a function of $\exp[-\|z - z'\|_2^2/2]$ and $S^L(x, x')$.*

*Proof.* We note that:

$$\Sigma^0(p(z), p(z')) = \frac{1}{k} p(z)^\top p(z) = \frac{1}{k} \sum_{i=1}^k \cos(W_i^1 z + b_i^1) \cos(W_i^1 z' + b_i^1) \tag{64}$$

By Thm. 1 in [7], we have:

$$\lim_{k \to \infty} \Sigma^0(p(z), p(z')) = \exp[-\|z - z'\|_2^2/2]/2 \tag{65}$$

which is a function of $\exp[\|z - z'\|_2^2]$ as desired. $\qquad\square$

We make use of the following lemma in the proof of Thm. 3.

**Lemma 6.** *Recall the parametrization of the implicit network:*

$$\begin{cases} g_i^l := g^l(z_i; v) = \sqrt{\frac{1}{m_{l-1}}} f^l(x_i; w) \cdot a_i^{l-1} \\ a_i^l := a^l(z_i; v) = \sqrt{2} \cdot \sigma(g_i^l) \end{cases} \quad \text{and} \quad a_i^0 := z_i \tag{66}$$

*For any pair $u_i = \{u_i\}$, we denote:*

$$P_i^{l_1 \to l_2} = \prod_{l=l_1}^{l_2 - 1} \left( \sqrt{\frac{2}{m_l}} V^{l+1}(x_i; w) \cdot Z^l(z_i) \right) \quad \text{and} \quad Z^l(z) = \text{diag}(\dot\sigma(g^l(z))) \tag{67}$$

*It holds that:*

1. $P_i^{l_1 \to l_2}(P_j^{l_1 \to l_2})^\top \xrightarrow{d} \prod_{l=l_1}^{l_2 - 1} \dot\Sigma^l(u_i, u_j) I.$

2. $\frac{\partial h(u_i, w)}{\partial v} \cdot \frac{\partial^\top h(u_j, w)}{\partial v} \xrightarrow{d} \sum_{l=0}^{H-1} \left( \Sigma^l(u_i, u_j) \prod_{h=l+1}^{H-1} \dot\Sigma^l(u_i, u_j) \right).$

*where the limits are taken with respect to $m, n \to \infty$ sequentially.*

*Proof.* We have:

$$\begin{aligned} &P_i^{l_1 \to l_2}(P_j^{l_1 \to l_2})^\top \\ =&P_i^{l_1 \to l_2 - 1} \frac{2}{m_{l_2 - 1}} V^{l_2}(x_i; w) \cdot Z^{l_2 - 1}(z_i) Z^{l_2 - 1}(z_j) V^{l_2}(x_j; w)^\top (P_j^{l_1 \to l_2 - 1})^\top \end{aligned} \tag{68}$$

Note that it holds that when $m, n \to \infty$ sequentially, we have:

$$\begin{aligned} &\frac{2}{m_{l_2 - 1}} V^{l_2}(x_i; w) \cdot Z^{l_2 - 1}(z_i) Z^{l_2 - 1}(z_j) V^{l_2}(x_j; w)^\top \\ &\xrightarrow{d} \sqrt{2} \mathop{\mathbb{E}}_{(u,v) \sim \mathcal{N}(0, \Lambda^{l_2})} [\dot\sigma(u)\dot\sigma(v)] I = \dot\Sigma^{l_2}(u_i, u_j) I \end{aligned} \tag{69}$$

Applying the above recursively proves the first claim. Using the first claim, along with the derivation of the neural tangent kernel (see [1]) proves the second claim. $\qquad\square$

**Theorem 3** (Hyperkernel convergence at initialization and composition). *Let* $h(u; w) = g(z; f(x; w))$ *be a hypernetwork. Then,*

$$\mathcal{K}^h(u, u') \xrightarrow{p} \Theta^h(u, u') \tag{70}$$

*where:*

$$\Theta^h(u, u') = \Theta^f(x, x') \cdot \Theta^g(u, u') \tag{71}$$

*such that:*

$$\mathcal{K}^f(x, x') \xrightarrow{p} \Theta^f(x, x') \cdot I \text{ and } \mathcal{K}^g(u, u') \xrightarrow{p} \Theta^g(u, u', S^L(x, x')) \tag{72}$$

*moreover, if $w$ evolves throughout gradient flow, we have:*

$$\left.\frac{\partial \mathcal{K}^h(u, u')}{\partial t}\right|_{t=0} \xrightarrow{p} 0 \tag{73}$$

*where the limits are taken with respect to $m, n \to \infty$ sequentially.*

*Proof.* Recalling that $v = vec(f(x)) = [vec(V^1), ..., vec(V^H)]$, concatenated into a single vector of length $\sum_{l=0}^{H-1} m_l \cdot m_{l+1}$. The components of the inner matrix $\mathcal{K}^f(x, x')$ are given by:

$$\mathcal{K}^f(x, x')(i, j) = \sum_{l=1}^{L} \left\langle \frac{\partial v_i(x)}{\partial w^l}, \frac{\partial v_j(x')}{\partial w^l} \right\rangle \tag{74}$$

and it holds that in the infinite width limit, $\mathcal{K}^f(x, x')$ is a diagonal matrix:

$$\mathcal{K}^f(x, x') \xrightarrow{d} \Theta^f(x, x') \cdot I \tag{75}$$

By letting the widths $n$ and $m$ tend to infinity consecutively, by Lem. 6, it follows that:

$$\frac{\partial h(u; w)}{\partial v} \cdot \frac{\partial^\top h(u'; w)}{\partial v} \xrightarrow{d} \Theta^g(u, u', S^L(x, x')) \tag{76}$$

Since $\mathcal{K}^f(x, x') = \frac{\partial f(x; w)}{\partial w} \cdot \frac{\partial^\top f(x'; w)}{w}$ converges to the diagonal matrix $\Theta^f(x, x') \cdot I$, the limit of $\mathcal{K}^h(u, u')$ is given by:

$$
\begin{aligned}
\mathcal{K}^h(u, u') =& \frac{\partial g(z; f(x; w))}{\partial f(x; w)} \cdot \frac{\partial f(x; w)}{\partial w} \cdot \frac{\partial^\top f(x'; w)}{w} \cdot \frac{\partial^\top g(z'; f(x'; w))}{\partial f(x'; w)} \\
=& \frac{\partial h(u; w)}{\partial v} \cdot \frac{\partial f(x; w)}{\partial w} \cdot \frac{\partial^\top f(x'; w)}{w} \cdot \frac{\partial^\top h(u'; w)}{\partial v} \\
\xrightarrow{d}& \Theta^f(x, x') \cdot \Theta^g(u, u', S^L(x, x'))
\end{aligned} \tag{77}
$$

where we used the results of Lem. 6.

Next, we would like to prove that $\left.\frac{\partial \mathcal{K}^h(u,u')}{\partial t}\right|_{t=0} = 0$. For this purpose, we write the derivative explicitly:

$$\frac{\partial \mathcal{K}^h(u, u')}{\partial t} = \frac{\partial h(u; w)}{\partial w} \cdot \frac{\partial}{\partial t}\frac{\partial^\top h(u'; w)}{\partial w} + \frac{\partial}{\partial t}\frac{\partial h(u; w)}{\partial w} \cdot \frac{\partial^\top h(u'; w)}{\partial w} \tag{78}$$

We notice that the two terms are the same up to changing between the inputs $u$ and $u'$. Therefore, with no loss of generality, we can simply prove the convergence of the second term. We have:

$$
\begin{aligned}
& \frac{\partial}{\partial t}\frac{\partial h(u; w)}{\partial w} \cdot \frac{\partial^\top h(u'; w)}{\partial w} \\
=& \left[\frac{\partial}{\partial t}\left(\frac{\partial h(u; w)}{\partial f(x; w)} \cdot \frac{\partial f(x; w)}{\partial w}\right)\right] \cdot \frac{\partial^\top h(u'; w)}{\partial w} \\
=& \left[\frac{\partial h(u; w)}{\partial f(x; w)\partial t} \cdot \frac{\partial f(x; w)}{\partial w} + \frac{\partial h(u; w)}{\partial f(x; w)} \cdot \frac{\partial f(x; w)}{\partial w \partial t}\right] \cdot \frac{\partial^\top h(u'; w)}{\partial w} \\
=& \frac{\partial h(u; w)}{\partial f(x; w)\partial t} \cdot \frac{\partial f(x; w)}{\partial w} \cdot \frac{\partial^\top h(u'; w)}{\partial w} + \frac{\partial h(u; w)}{\partial f(x; w)} \cdot \frac{\partial f(x; w)}{\partial w \partial t} \cdot \frac{\partial^\top h(u'; w)}{\partial w}
\end{aligned} \tag{79}
$$

We analyze each term separately.

**Analyzing the first term** By substituting $\frac{\partial}{\partial t} = -\mu \nabla_w c(w) \frac{\partial^\top}{\partial w} = -\mu \nabla_w c(w) \frac{\partial^\top f}{\partial w} \frac{\partial^\top}{\partial f}$, we have:

$$\frac{\partial h(u;w)}{\partial f(x;w)\partial t} \cdot \frac{\partial f(x;w)}{\partial w} \cdot \frac{\partial^\top h(u';w)}{\partial w}$$

$$= -\mu \nabla_w c(w) \frac{\partial^\top f(x;w)}{\partial w} \cdot \frac{\partial^2 h(u;w)}{\partial f(x;w)\partial f(x;w)} \cdot \frac{\partial f(x;w)}{\partial w} \cdot \frac{\partial^\top f(x';w)}{\partial w} \cdot \frac{\partial^\top h(u';w)}{\partial f(x';w)}$$

$$= -\mu \nabla_w c(w) \frac{\partial^\top f(x;w)}{\partial w} \frac{\partial^2 h(u;w)}{\partial f(x;w)\partial f(x;w)} \mathcal{K}^f(x,x') \cdot \frac{\partial^\top h(u';w)}{\partial f(x';w)}$$

$$= -\mu \sum_{i=1}^N \frac{\partial \ell(h(u_i;w),y_i)}{\partial h(u_i;w)} \cdot \frac{\partial h(u_i;w)}{\partial f(x;w)} \cdot \mathcal{K}^f(x,x_i) \cdot \frac{\partial^2 h(u;w)}{\partial f(x;w)\partial f(x;w)} \cdot \mathcal{K}^f(x,x') \cdot \frac{\partial^\top h(u';w)}{\partial f(x';w)}$$

$$(80)$$

It then follows:

$$\lim_{n\to\infty} \frac{\partial h(u;w)}{\partial f(x;w)\partial t} \cdot \frac{\partial f(x;w)}{\partial w} \cdot \frac{\partial^\top h(u';w)}{\partial w}$$

$$= -\mu \sum_{i=1}^N \ell_i \cdot \Theta^f(x,x_i) \cdot \Theta^f(x,x') \lim_{n\to\infty} \frac{\partial h(u_i;w)}{\partial f(x_i;w)} \cdot \frac{\partial^2 h(u;w)}{\partial f(x;w)\partial f(x;w)} \cdot \frac{\partial h(u';w)}{\partial f(x';w)}$$

$$(81)$$

We notice that:

$$\lim_{n\to\infty} \frac{\partial^\top h(u_i;w)}{\partial f(x_i;w)} \cdot \frac{\partial^2 h(u;w)}{\partial f(x;w)\partial f(x;w)} \cdot \frac{\partial h(u';w)}{\partial f(x';w)}$$

$$= \sum_{l_1,l_2} \lim_{n\to\infty} \left\langle \frac{\partial^2 h(u;w)}{\partial f^{l_1}(x;w)\partial f^{l_2}(x;w)}, \frac{\partial h(u_i;w)}{\partial f^{l_1}(x_i;w)} \otimes \frac{\partial h(u';w)}{\partial f^{l_2}(x';w)} \right\rangle$$

$$:= \sum_{l_1,l_2} \mathcal{T}_m^{l_1,l_2}(u,u_i,u')$$

$$(82)$$

We recall that $f^l(x;w)$ converges to a GP (as a function of $x$) as $n \to \infty$ [4]. Therefore, $\mathcal{T}_m^{l_1,l_2}(u,u_i,u')$ are special cases of the terms $\mathcal{T}_{n,i,d}^{l,i,d}$ (see Eq. 7) with weights that are distributed according to a GP instead of a normal distribution. In this case, we have: $k = 2, d = d_1 = \cdots = d_k = 1$, the neural network $f^1$ is replaced with $h$, the weights $W^l$ are translated into $f^l(x;w)$. We recall that the proof of Lem. 3 showing that $\mathcal{T}_{n,i,d}^{l,i,d} = \mathcal{O}_p(1/n^{k-1})$ is simply based on Lem. 2. Since Lem. 6 extends Lem. 2 to our case, the proof of Lem. 3 can be applied to show that $\mathcal{T}_m^{l_1,l_2}(u,u_i,u') \sim 1/m$.

**Analyzing the second term** We would like to show that for any $m > 0$, we have:

$$\frac{\partial h(u;w)}{\partial f(x;w)} \cdot \frac{\partial f(x;w)}{\partial w \partial t} \cdot \frac{\partial^\top h(u';w)}{\partial w} \xrightarrow{d} 0 \qquad (83)$$

as $n \to \infty$. Since $\frac{\partial w}{\partial t} = -\mu \nabla_w c(w)$, we have:

$$\frac{\partial h(u;w)}{\partial f(x;w)} \cdot \frac{\partial f(x;w)}{\partial w \partial t} \cdot \frac{\partial^\top h(u';w)}{\partial w}$$

$$= -\mu \cdot \frac{\partial h(u;w)}{\partial f(x;w)} \cdot \nabla_w c(w) \cdot \frac{\partial^2 f(x;w)}{\partial w^2} \cdot \frac{\partial^\top h(u';w)}{\partial w}$$

$$= -\mu \cdot \frac{\partial h(u;w)}{\partial f(x;w)} \cdot \nabla_w c(w) \cdot \frac{\partial^2 f(x;w)}{\partial w^2} \cdot \frac{\partial^\top f(x';w)}{\partial w} \cdot \frac{\partial^\top h(u';w)}{\partial f(x;w)}$$

$$(84)$$

In addition, we have:

$$\nabla_w c(w) = \sum_{i=1}^N \frac{\partial \ell(h(u_i;w),y_i)}{\partial h(u_i;w)} \cdot \frac{\partial h(u_i;w)}{\partial w} \qquad (85)$$

We note that $\frac{\partial \ell(h(u_i;w),y_i)}{\partial h(u_i;w)}$ converges in distribution as $m,n \to \infty$. Therefore, we can simply analyze the convergence of:

$$\sum_{i=1}^N \frac{\partial h(u;w)}{\partial f(x;w)} \cdot \frac{\partial h(u_i;w)}{\partial w} \cdot \frac{\partial^2 f(x;w)}{\partial w^2} \cdot \frac{\partial^\top f(x';w)}{\partial w} \cdot \frac{\partial^\top h(u';w)}{\partial f(x;w)} \qquad (86)$$

Since $N$ is a constant, it is enough to show that each term converges to zero. We have:

$$
\frac{\partial h(u;w)}{\partial f(x;w)} \cdot \frac{\partial h(u_i;w)}{\partial w} \cdot \frac{\partial^2 f(x;w)}{\partial w^2} \cdot \frac{\partial^\top f(x';w)}{\partial w} \cdot \frac{\partial^\top h(u';w)}{\partial f(x;w)}
$$

$$
= \frac{\partial h(u;w)}{\partial f(x;w)} \cdot \frac{\partial h(u_i;w)}{\partial f(x_i;w)} \cdot \frac{\partial f(x_i;w)}{\partial w} \cdot \frac{\partial^2 f(x;w)}{\partial w^2} \cdot \frac{\partial^\top f(x';w)}{\partial w} \cdot \frac{\partial^\top h(u';w)}{\partial f(x;w)} \tag{87}
$$

$$
= \sum_{l,j,k} \frac{\partial h(u;w)}{\partial f(x;w)_l} \cdot \frac{\partial h(u_i;w)}{\partial f(x_i;w)_j} \cdot \frac{\partial f(x_i;w)_j}{\partial w} \cdot \frac{\partial^2 f(x;w)_l}{\partial w^2} \cdot \frac{\partial^\top f(x';w)_k}{\partial w} \cdot \frac{\partial^\top h(u';w)}{\partial f(x;w)_k}
$$

where $f(x;w)_j$ is the $j$'th output of $f$ over $x$. In addition, the summation is done over the indices of the corresponding tensors. We note that for any $m > 0$, the number of indices $l, j, k$ is finite. We would like to show that each term in the sum tends to zero as $n \to \infty$. We can write:

$$
\frac{\partial f(x_i;w)_j}{\partial w} \cdot \frac{\partial^2 f(x;w)_l}{\partial w^2} \cdot \frac{\partial^\top f(x';w)_k}{\partial w} = \left\langle \frac{\partial^2 f(x;w)_l}{\partial w^2}, \frac{\partial f(x_i;w)_j}{\partial w} \otimes \frac{\partial f(x';w)_k}{\partial w} \right\rangle \tag{88}
$$

By Lem. 3, the term in Eq. 88 tends to zero as $n \to \infty$. In addition, it is easy to see that $\frac{\partial h(u;w)}{\partial f(x;w)_l}$, $\frac{\partial h(u_i;w)}{\partial f(x_i;w)_j}$ and $\frac{\partial^\top h(u';w)}{\partial f(x;w)_k}$ converge to some random variables. Therefore, for any fixed $m > 0$, the above sum converges to zero as $n \to \infty$. □