[Reviews · NeurIPS 2020]

Review 1

Summary and Contributions: The manuscript analyzes the hypernetworks and considers the case where the meta-network and the primary-network is infinitely wide. The authors show it is necessary that both these components are infinitely wide in order to obtain a linear optimization problem. From this observation, they derive the neural tangent kernel of hypernetwork (the hyperkernel)

Strengths: The paper is well written and pleasant to read and it proposes and interesting and novel approach based on the hypernetwork. Moreover, the authors propose a clear and interesting analysis of the impact of applying the concept of infinity width in the hypernetworks

Weaknesses: The main weakness is related to the experimental section where an empirical comparison between the hyperkernel and state-of-the-art models in a complex scenario is missing. The used dataset (MNIST) is useful to analyze the proposed model, but the faced tasks are not complex enough to assess the strengths and the weaknesses of the proposed method. Another problem is that the mathematical notation in some points is not completely clear, (e.g in theorem 3) and a more detailed description should be inserted. In section 3.2 the authors state: “ Since typically, the meta-network f is much larger than the primary network”, but in Ha, David, Andrew Dai, and Quoc V. Le. "Hypernetworks." it is suggested to do the exact opposite. Therefore the authors should motivate why in this particular case the meta-network should be larger than the primary-network. Minor comments: In equation (1) I guess that f^l has to be substituted with y_l, otherwise please provide an explanation about f^l. In figure 4 the scales used for accuracy in the 2 graphs are different. To make them more readable (and to simplify the comparison) please use the same scale in both graphs.

Correctness: The proposed method and the analysis seem correct.

Clarity: The paper is clear and well written

Relation to Prior Work: The provided literature review is complete and exhaustive

Reproducibility: Yes

Additional Feedback: --- Update after author response --- I have read the authors' rebuttal and other reviews. I want to thank the authors for exhaustive responses that address many issues raised by me and the other reviews. I think that the paper is very interesting, well written, and very relevant to the Hypernetwork research field. Therefore I decided to keep my original score.


Review 2

Summary and Contributions: The paper follows a recent vein of analyzing the NTK for different neural network architectures and, in particular, studies the kernel dynamics of hypernetworks. It shows that for a fixed-width primary network and infinite-width 'meta-network', the limiting behavior is not characteristic of the kernel regime. On the other hand, taking infinite-widths for both networks yields lazy training dynamics. Experimental results compare the performance of hypernetworks and 'hyperkernels' on small-scale tasks.

Strengths: The paper is well written and the theoretical results are interesting/relevant.

Weaknesses: I believe the main weaknesses are: poor/artificial connection to hypernetworks (the analyzed model, the nomenclature and experiments are mostly inconsistent with prior work), and unconvincing/confusing experimental setup. More details follow below. - Connection to Hypernetworks The paper is clearly focused on analyzing hypernetworks, but at the same time does not consider optimization dynamics induced by many (if not most) uses of hypernetworks. The adopted nomenclature is confusing and non-standard: prior work (e.g. references [6,21,28,35] of the submission) refers to the 'meta-network' (the model that generates weights) as the hypernetwork, while the submission adopts the 'meta-network' term and refers to the two models (meta+primary nets) as the 'hypernetwork'. Moreover, the studied dynamics here consider the model given in Eq. 2, where the meta-network parameters are optimized to minimize some empirical loss (Eq. 3) evaluated on a set of points, where each point u_i defines an input for the primary net and for the meta-network -- it is unclear why such training setup was adopted since it does not match nor include typical uses of hypernetworks. While I believe that understanding optimization dynamics of the meta-network parameters is interesting, there is little insight gained if the analysis stops there. Considering the references in the submission: [6] - A hypernetwork receives an architecture descriptor as input and generates weights for its underlying computational graph, and is trained to minimize the loss of the produced network. Once the hypernetwork is trained, the hypernet input (descriptor) is optimized (via sampling) to perform architecture search. [21] - A hypernetwork receives training hyperparameters as input and is trained to match the best-response function for these hyperparams (i.e. output weights that minimize the loss defined by the hyperparams). Once the hypernetwork is trained, the hypernet inputs are optimized to perform hyperparameter search. [28] - A hypernetwork receives task-specific embeddings as input and is trained to produce weights suitable for the task. The hypernetwork inputs (task embeddings) are trained jointly with its parameters. [35] - Closely related to [6], except inputs are node embeddings computed from the architecture's computational graph. Note that all cases involve optimizing the hypernetwork's inputs once its parameters have been trained, which is indeed the standard and most successful use of hypernetworks. Except for [28], the inputs are optimized while keeping the trained parameters fixed. On the other hand, [28] uses hypernetworks to decrease the number of parameters involved in continual learning, hence it is arguably a poor example of a hypernetwork application when studying infinite-width training dynamics of these models. It seems that the adopted training model only includes the use in [28], where the goal is not finding optimal inputs for a trained hypernetwork, but actually minimizing an explicit empirical loss where hypernetwork inputs are given (in that specific case, task indicators). A minimally complete pipeline would be to not only study optimization dynamics of the hypernetwork parameters, but also the dynamics when optimizing the hypernetwork inputs once its parameters have been trained and fixed. Lastly, the experimental setup brings yet another inconsistency with prior work on hypernetworks: none of the experiments characterizes a meaningful and standard use of hypernetworks. More details on that further below. - Experimental setup For the image completion and inpainting, the problem was set up as a regression problem from pixel coordinates to pixel intensities, with the perturbed image given as additional input to the hypernetwork -- there is no motivation given on why the problem was framed in such an artificial manner, and if the goal was to set up the task as a learning a conditional mapping, it would be considerably cleaner and more natural to have the digit number (label) given as input to the meta-network and the corrupted image as input to the primary network, whose goal would be to output a full reconstruction. Similarly for the rotated MNIST experiment, feeding the meta-network with the original image and the primary-net with the rotated one seems arbitrary and artificial. Note that neither of the two tasks seem natural for hypernetworks, and it is unclear whether a standard network would actually perform worse here.

Correctness: The paper seems correct.

Clarity: Yes.

Relation to Prior Work: The discussion on prior work on NTK and GP is clear and seems complete. As mentioned above, the connection to prior work on hypernetworks is a bit fuzzy since the models and uses are mostly inconsistent (which is never mentioned in the paper).

Reproducibility: Yes

Additional Feedback: The authors claim that the hyperkernel has a computational advantage, (L208), but it is unclear what this statement means. The fact that the hyperkernel has a compositional structure and the meta-network term can be re-used for distinct data points that share the same meta-network input, not surprisingly, bears similarity to training and inference of hypernetworks, where weights only need to be produced once (i.e. only a single pass of the meta-network) for a set of data points that share meta-network inputs. Note that this is standard practice in hypernetwork implementations, hence the statement that 'hyperkernels' have an advantage is vague and confusing, and some clarification would be appreciated. --- Update after author response --- After reading the author's rebuttal, I believe there are still major concerns in terms of nomenclature, analyzed model, and prior work. The rebuttal attempts to support the model analyzed in the paper (Eq 2 & 3) and the experimental setting by referring to a new set of papers (A,B,C,D,E in the rebuttal), which, in my opinion, end up raising additional concerns. Out of these, A, B, and D either don't support the model adopted by the authors at all (the dynamic filters in A, if seen as a hypernetwork, would require multiple connections between f and g) or use a similar model but fall in the category of implicit representation learning (implicit shape representation in B, implicit image representation in E), and not hypernetworks (in terms of goals, technique, and target academic community). I feel that adopting a model that is almost exclusively adopted in papers that perform implicit representation learning and referring to them as hypernetworks to be quite problematic, as it spreads confusion among readers and the community. The fact that the paper exclusively discusses standard hypernetwork papers (L59 onwards), where the goal is typically to optimize the hypernetwork's inputs post-training, and not to simply perform ERM, while not discussing implicit representation learning papers whose model & training actually has some similarity to their adopted ones, is particularly worrying. The rebuttal raised additional concerns and, after reading it carefully, I have updated my score to reflect this. I believe the paper would be heavily improved by adopting the correct nomenclature, discussing papers that use a similar model (e.g. B and E), and being upfront about the limitations of the theoretical results, for example not being able to capture the settings where hypernetworks are typically used. Without these revisions I cannot recommend its acceptance.


Review 3

Summary and Contributions: In the context of hyper-networks the paper addresses the research question about their behaviour in the over-parameterized regime when randomly initialized. Theoretical results, obtained by extending already known results for standard networks, show that when both the meta and primary networks are infinitely wide, the computed function exhibits Gaussian Process and Neural Tangent Kernel behaviours. A new tighter upper bound on the convergence rate for standard networks is also provided as a by-product. Some empirical assessment are presented as well.

Strengths: - The covered topic is timely and relevant for the NeurIPS community. - The presented theoretical results seem to be original and of interest. - Empirical assessment seems to provide support for the theoretical results.

Weaknesses: - The reported results seem to be partially derivative: extension to hyper-networks of results already presented in the literature for standard networks. - The case with finite width for f and infinite width for g is not discussed: it would have provided a complete treatment of the topic. - Presentation could be improved, first of all by removing typos (see additional comments), and then by providing more background on NTK and GP.

Correctness: - The presented approach seems to be sound. - Empirical demonstration could have been more extended. Moreover, no variance is reported in the experimental results in Table 1 and 2, so it is not clear how much the experimental assessment can be considered sound.

Clarity: The paper is reasonably clear although not easy to read for a non expert of the topic. I guess background material on NTK and GP would be of help to better follow the arguments. Concerning the experimental assessment, I am not sure that the authors provided all the information needed to reproduce the discussed results. In this respect: Figure 2(b) is missing reference to f and so it is not clear what it is showing; moreover, Tables 1 and 2 do not provide any variance in the reported results, so it is not clear if the reported results are obtained by a single trial or are the average of several trials (and variances are not reported); finally, authors should discuss why the HN with large width seems to outperform the HK (variances in the experimental results would have provided support for a possibile explanation).

Relation to Prior Work: The paper seems to refer the main contributions in the field and to explain the difference with respect to them.

Reproducibility: No

Additional Feedback: Here are some typos that need to be fixed and few suggestions to improve presentation. line 73: "In [10] they theoretically" -> In [10] the authors theoretically" line 79: "We begin be defining" -> "We begin by defining" eq. (19): f^l should be y^l line 84: "to be the a piece-wise linear function" -> "to be a piece-wise linear function" line 93: "R^{n_0 × m_0}" -> "R^{n_0 + m_0}" line 97: "w ∈ R^N stands" -> N is already used for the number of training examples line 109: "U[N]" is not defined; moreover, here a double index (j_t) is introduced with no real explanation, so it would be better to spend more words to explain it line 125: "Dynamics Of Hypernetworks" -> "Dynamics of Hypernetworks" line 131: "as training of of f"-> "as training of f" lines 137-9: I assume "j" is the index of the example presented just before i, and that lead to a change in the weights: this should be said explicitly. If this is not the case, then I am unable to follow the argument; moreover, the full stop before the "where" in eq. (4) should be a comma. line 140: "Previous work have shown" -> "Previous work has shown" or "Previous works have shown" line 180: "the parameters contain randomly initialized rank 3 tensors all of whose dimensions" -> the authors should help the less expert reader to understand why rank 3 tensors are involved. Caption Figure 2: "Convergence to a deterministic kernel is observed only when both f and g are wide." -> in Figure 2(b) there is no reference to f, so the statement is not clear at all. line 209: "hyperkernel serves a a functional" -> "hyperkernel serves a functional" Figures 1 and 3 are not cited in the main text; Figure 1 can be removed, since it is not needed. ========== I have read the author rebuttal and the other reviews. Notwithstanding the issues of some reviewers on inconsistent nomenclature and notation, experimental design, and lack of effectiveness, I still think the paper, in its final revised form, can give a useful contribution to the field, so I decided to keep my original score.


Review 4

Summary and Contributions: The authors consider hypernetworks (HN) composed of two networks (meta-networks and primary networks) with respect to neural tangent kernel (NTK) framework. Their main claims are two-fold: (1) To study the characteristics of HNs and the relation to Gaussian Processes (GP) and NTK when those widths are wide enough. The NTK of HNs (hyperkernel) is their main focus. (2) Hyperkernels have large representational power and they are advantageous for the meta-learning tasks with low size of datasets. They verify experimentally the convergence conditions of the hyperkernels. And demonstrated are the performance of exemplar applications such as image completion/inpainting and MNIST rotation prediction.

Strengths: For HNs, the authors derive the convergence conditions for linear approximation of HNs, and the relation to GP and NTK thoroughly. Eventually, the hyperkernels are well-defined from the above derivations. The hyperkernels successfully are applied to some computer vision tasks.

Weaknesses: The authors claim that the hyperkernels are good for meta learning problems. However, the problems in Section 5 does not include typical examples for them. It seems that the comparative study is not fairly designed, and it is not easy to check due to lack of the setting detail for the original hypernetworks. It is not trivial. How to give context points for the hypernetworks?

Correctness: Most claims for theory are thoroughly derived. (See Strengths) But, there are some inconsistent experiment settings and the results with the manuscript. See Weakness above.

Clarity: I feel confused whether the purpose of this paper is theory of the hyperkernels or the application (as functional prior when there is little data). Most parts are assigned for the theory. However, the experiments are mostly assigned for the applications (not so appropriate for their main claims). I think that it is better to keep the purpose and tone of writing consistent throughout the manuscript. Some formulae has not-defined terms. (see additional feedback)

Relation to Prior Work: Well surveyed to understand the position of this work for theory of the hyperkernels.

Reproducibility: No

Additional Feedback: Theoretically, the experiments for meta-learning or few-shot learning would be more persuasive. Practically, it would be powerful to apply real-world problems like the following, if possible. - Klocek et al., Hypernetwork functional image representation, ICANN 2019, (https://arxiv.org/abs/1902.10404) The kappa should be explained in Formula (4) * Minors the right dot of Kappa is necessary in Formula (4)? line 133: of of line 209: a a line 215: then --> than? Formula (15): z(0) ? z(theta)? After Rebuttal ----------------- Considering all reviews, I think that this manuscript has common issues among most reviewers: (1) inconsistent nomenclature and notation, (2) experimental designs are not appropriate to support main claims, and (3) the effectiveness of the proposed methods is not validated eventually. The rebuttal partially deals with my concerns, and I think that it does not resolve the common main issues properly. Thus, I keep my score as it is.

[Author Response · NeurIPS 2020]

**R1:** **The experimental setup** of our paper is designed to both corroborate our theoretical results which are non-trivial,
while also demonstrating the possible application of the hypernetwork induced prior in some practical use cases. We
concede that our experiments are by no means thorough enough to consider the hypernetwork kernels as a go-to
algorithm for image completion tasks, however we feel they do serve their purpose demonstrating the usefulness of
hypernetworks induced priors, explaining the validity and inductive bias of the architecture. **Size of** $f$**:** While in some
of the hypernetworks in the literature $f$ is not very large, this is the case in many of the recent networks, e.g., [B,C,D,E].
**R2:** **We accept the suggested terminology** and would use "hypernet" to refer to network $f$ only. **There are two types**
**of hypernetwork** architectures (including the hypernetwork $f$ and the primary network $g$). Type A has a much larger
network $f$ than $g$, and the input to $f$ is larger than the input to $g$. Type B has a smaller $f$ and larger $g$. In the references
that the reviewer mentioned, where one optimizes the input of $f$, it is more natural to use type B (smaller inputs).
However, type A is at least as prevalent in the literature as type B. Examples of type A include cases where $g$ is a
single convolutional layer in a deeper network [17,15,5,A]. This is also very prominent in recent work in which the
perceptual task is done by a resnet $f$ and the solution is parameterized by a small network $g$ [B,C,D,E]. In such cases, $f$
observes the entire context, while $g$ is a local model, see also [F]. Our analysis holds also for type B networks, given
that the network $f$ is wide enough to be approximated by a GP. However, since type B networks are used to find optimal
hyperparameters, they require training by definition, and NTKs, which are studied at initialization, are less relevant.
Indeed, there is currently no theoretical machinery for understanding the dynamics of optimizing the input with a fixed,
trained network in the NTK regime. Our work here presents a first step at a new understanding of hypernetworks through
the new machinery of NTK, which covers the type A scenario. We hope to extend this analysis to type B scenarios in
the future as well by using the recent Tensor Programs framework. However, this is out of the scope of this contribution.
**The MNIST experiment** is a typical type A hypenet setting. The perceptual input is processed by $f$, and $g$ is a model
of the "scene". It directly follows [B,E] (E is a paper R4 pointed to as an example for realistic settings). The reviewer's
suggestion to condition $f$ on the digit label is equivalent to learning 10 different denoising networks, which is not
utilizing the full power of hypernets. **The computational advantage (L 208)** is meaningful when compared to other
kernel methods. From the composition of the hyperkernel (Eq. 12), $\Theta^{\hat{f}}(x, x')$ can be evaluated separately for all pairs
$x, x'$, instead of evaluating kernel values for all pair of tuples $(x, z), (x', z')$ when considering other kernel methods.
When $f$ is a convnet, this can represent a significant reduction in computational cost. **R3:** **Our theoretical results are**
**non-trivial.** In particular, Thm. 1 provides the asymptotic behavior of high order NTK terms which hold for ReLU
hypernetworks, as well as regular ReLU MLPs. Our contribution here is both a technical novelty (in the proof) and the
significance of the final result. On the technical level, as noted in remark 1 and in L 280-283, we have proven (and
improved upon) a conjecture on the asymptotic rates of various correlation functions arising in neural network dynamics
(see [5]). As for the result itself, we are the first to arrive at these tight bounds which relate to both hypernetworks and
MLPs. Thms. 2 and 3 describe the conditions in which GP behavior emerges in hypernetworks (again nontrivial), and
describe the composition of the GP and NTK kernels. We feel these theoretical results are of interest to the community.
**The case of a finite** $f$ **and an infinite** $g$ is left for future work. Note that an infinite $g$ would require $f$ to output an
infinite number of parameters. We do discuss the case of both $f$ and $g$ being infinite in Sec. 4. **Reporting variance** we
regret not reporting error bars, which will be added. The results were averaged over 10 different training and test splits.
**HN outperforms HK in Tab. 2** For small data regimes, the HK outperforms,
while for larger datasets, the HN is better. This is consistent with prior observa-
tions that kernel methods with NTK tend to outperform in low data regimes. In
**Fig. 2**, the input of $f$ is depicted in Row 2, the input of $g$ is a pixel coordinate.
The output of $g$ is the pixel intensity in the corresponding coordinate. **Typos**
We apologize for the typos and would provide more background on NTK and
GP. **R4:** The paper is theoretically driven. **The experiment setup** is very sim-
ilar to that of the mentioned paper (arXiv 1902.10404) only done on MNIST.
To demonstrate this, Fig. I has interpolation results similar to that paper by in-
terpolating between $[\Theta^h(u, u_1), .., \Theta^h(u, u_N)]$ and $[\Theta^h(v, u_1), .., \Theta^h(v, u_N)]$
for two images $u$ and $v$ in Eq. 16. **We regret the lack of details** in the exper-
imental section. The hypernet $f$ in our setup is a convolutional neural network
operating on sparse images containing the context points, similarly to [G].

Figure I: Interpolation between 7 ->2 and 5 ->6 using hyperkernel (rows 1 ,3) and hypernetwork (rows 2,4). Both methods used merely 200 samples for training. The hypernetwork trained with sgd clealy underperforms in this low data regime

REFERENCES: **[A]** Wu et al. Pay Less Attention with lightweight and Dy-
namic Convolutions. ICLR 2019. **[B]** Littwin et al. Deep Meta Functionals for
Shape Representation. ICCV, 2019. **[C]** Rotman et al. Electric Analog Circuit
Design with Hypernetworks and a Differential Simulator. ICASSP, 2020. **[D]**
Bergman et al. Implicit neural representations with periodic activation func-
tions. arXiv:2006.09661, 2020. **[E]** Klocek et al. Hypernetwork functional image representation. ICANN, 2019. **[F]**
Nachmani et al. Hyper-Graph-Network Decoders for Block Codes. NeurIPS, 2019. **[G]** Sitzmann et al. Implicit Neural
Representations with Periodic Activation Functions. arXiv:2006.09661, 2020.


[Meta-Review · NeurIPS 2020]

The authors present a theoretical analysis of the asymptotic behavior of high order neural tangent kernel terms which holda for regular ReLU MLPs as well as ReLU hypenets (according to the terminology after author response period) . The results settle a previous conjecture on the asymptotic rates of various correlation functions arising in neural network dynamics. In particular, the results relate to both hype-nets and MLPs. Numerical illustration are provided in the paper. The reviewers appreciated the novelty of the proposed approach based on hypenets. They also noted that the topic is timely and relevant for the community and that the results presented answer important questions in the field. A reviewer commented that for hypenets 'the authors derive the convergence conditions for linear approximation of HNs, and the relation to GP and NTK thoroughly'. The reviewers, however, expressed concerns about 'nomenclature, analyzed model, and prior work'. The authors submitted a response to the reviewers' comments, as well as confidential comments to the area chair. After reading the response, updating the reviews, and discussion, the reviewers feel 'the paper would be heavily improved by adopting the correct nomenclature, discussing papers that use a similar model, and being upfront about the limitations of the theoretical results, for example not being able to capture the settings where hypernetworks are typically used'. The reviewers provided valuable hints on the directions for improvement towards the final version. We highly recommend to take the reviewers' suggestions into account while preparing the camera ready final version of the paper. The paper makes timely and relevant contributions the field, proving rigorous mathematical convergence results for the linear approximation of hype-nets, in relation with GPs and NTKs. This paper will likely be a reference in that area. Accept.